# Parallel processing of quickly and slowly mobilized reserve vesicles in hippocampal synapses

Juan Jose Rodriguez Gotor[1†], Kashif Mahfooz[2†], Isabel Perez-Otano[1], John F Wesseling[1]*

[1]Instituto de Neurociencias de Alicante CSIC-UMH, San Juan de Alicante, Spain; [2]Department of Pharmacology, University of Oxford, Oxford, United Kingdom

**Abstract** Vesicles within presynaptic terminals are thought to be segregated into a variety of readily releasable and reserve pools. The nature of the pools and trafficking between them is not well understood, but pools that are slow to mobilize when synapses are active are often assumed to feed pools that are mobilized more quickly, in a series. However, electrophysiological studies of synaptic transmission have suggested instead a parallel organization where vesicles within slowly and quickly mobilized reserve pools would separately feed independent reluctant- and fast-releasing subdivisions of the readily releasable pool. Here, we use FM-dyes to confirm the existence of multiple reserve pools at hippocampal synapses and a parallel organization that prevents intermixing between the pools, even when stimulation is intense enough to drive exocytosis at the maximum rate. The experiments additionally demonstrate extensive heterogeneity among synapses in the relative sizes of the slowly and quickly mobilized reserve pools, which suggests equivalent heterogeneity in the numbers of reluctant and fast-releasing readily releasable vesicles that may be relevant for understanding information processing and storage.

*For correspondence: john.wesseling@csic.es

†These authors contributed equally to this work

Competing interest: The authors declare that no competing interests exist.

## eLife assessment

This study addresses the long-standing question as to how different functional pools of synaptic vesicles are organized in presynaptic terminals to mediate different modes of neurotransmitter release. Based on imaging of active synapses with recycling synaptic vesicles labeled by FM-styryl dyes, the authors provide data that are compatible with the hypothesis that two separate reserve pools of vesicles – slowly vs. rapidly mobilizing – feed two distinct releasable pools – reluctantly vs. rapidly releasing. Overall, this study represents a **valuable** contribution to the field of synapse biology, specifically to presynaptic dynamics and plasticity. The authors' methodological approach of using bulk FM-styryl dye destaining as a readout of precise vesicle arrangements and pools in a population of functionally very diverse synapses has limitations. Consequently, the evidence that directly supports the authors' two-pool-interpretation of their data is **incomplete**, and alternative interpretations of the data remain possible.

## Introduction

Chemical synapses exhibit striking dynamic changes in connection strength during repetitive use. The changes are termed *short-term plasticity* or *frequency dynamics* and are thought to play an important role in how information is processed (*Tsodyks and Markram, 1997*; *Abbott and Regehr, 2004*; *Buonomano and Maass, 2009*). Multiple presynaptic vesicle trafficking mechanisms are involved, but

**Figure 1.** Three possible organizations for synaptic vesicle pools. (**A**) The predominant view currently seems to be that all pools are connected in series as depicted; *RRP* signifies readily releasable pool. (**B** and **C**) However, recent evidence indicates that vesicles within reluctant and fast-releasing subdivisions of the RRP can be released in parallel. (**D**) And, a separate line of evidence suggested that each vesicle within any subdivision of the RRP is associated with an autonomous reserve, implying that slowly and quickly mobilized reserves may also be processed in parallel, as depicted in Panel **C**.

the identity of the mechanisms, how they interact, and the implications for biological computation are not understood (*Neher, 2015*).

Most attempts at a detailed understanding begin with the premise that vesicles are segregated into multiple *pools*, including at least one readily releasable pool and one reserve pool. Readily releasable vesicles are thought to be docked to *release sites* embedded within the active zone of the plasma membrane of synaptic terminals, whereas reserves reside in the interior. Nevertheless, pools are typically not defined by morphological criteria but instead by the timing of *mobilization*, which is a general term for the full sequence of events that must occur before vesicles undergo exocytosis; for reserve vesicles, mobilization would include docking to release sites, biochemical priming, and catalysis of exocytosis.

Based on timing criteria, the readily releasable pool has been divided into *fast-* and *slow-releasing* subdivisions at a wide variety of synapse types; slow-releasing readily releasable vesicles are often termed *reluctant* (*Wu and Borst, 1999*; *Sakaba and Neher, 2001*; *Moulder and Mennerick, 2005*). Reserve pools are less studied, but have likewise been divided into quickly and slowly mobilized pools at some synapse types (*Neves and Lagnado, 1999*; *Rizzoli and Betz, 2005*; *Denker et al., 2011*).

A widespread premise has been that the pools are connected in a series where vesicles are transferred from slowly to quickly mobilized pools as diagrammed in *Figure 1A* (*Pieribone et al., 1995*; *Hilfiker et al., 1999*; *Denker and Rizzoli, 2010*; *Rothman et al., 2016*; *Miki et al., 2016*; *Doussau et al., 2017*; *Milovanovic et al., 2018*). If so, reluctant vesicles might be docked to the same type of release site as fast-releasing vesicles, but in an immature priming state that could then transition to

the mature, fast-releasing state. We refer to models with this premise as *homogeneous release site* models (e.g. *Lin et al., 2022*).

However, a growing body of evidence suggests that reluctant vesicles are not immature but are instead fully primed at release sites that are inherently inefficient at catalyzing exocytosis. First, to our knowledge, none of the homogeneous release site models proposed so far can account for a series of electrophysiological experiments at calyx of Held synapses where: (1) the reluctant subdivision was only minimally depleted during moderate stimulation that exhausted the fast-releasing subdivision; and (2) abruptly increasing the stimulation intensity then drove exocytosis of the remaining reluctant vesicles directly, without first transitioning to a fast-releasing state (*Mahfooz et al., 2016* and Figure 5—figure supplement 1 of *Raja et al., 2019*). And second, recent optical imaging and molecular studies have confirmed substantial differences between release sites within the same active zone at a wide variety of synapse types (*Hu et al., 2013*; *Müller et al., 2015*; *Böhme et al., 2016*; *Akbergenova et al., 2018*; *Maschi and Klyachko, 2020*; *Li et al., 2021*; *Karlocai et al., 2021*; *Gou et al., 2022*).

To account for the observations, one of the starting premises of our working model is that fast-releasing and reluctant subdivisions of the readily releasable pool operate in parallel, which could be as in *Figure 1B* if based solely on the studies referenced so far. *Figure 1C* depicts a different possibility, tested here, where quickly and slowly mobilized reserve pools are likewise arranged in parallel, and independently supply vesicles to corresponding fast-releasing and reluctant subdivisions of the readily releasable pool.

No such fully parallel arrangement has been proposed previously, at least not explicitly. However, an additional premise of our working model - based on separate studies, not related to distinctions between subdivisions of the readily releasable pool - is that each readily releasable vesicle is physically tethered to a short chain of reserve vesicles, which serve as replacements after the readily releasable vesicle undergoes exocytosis (*Figure 1D*; *Gabriel et al., 2011*; see *Wesseling et al., 2019* for supporting ultrastructural evidence).

If so, reserve vesicles chained to reluctant readily releasable vesicles would advance to the release site more slowly during continuous stimulation than reserve vesicles chained to fast-releasing vesicles because the reluctant vesicles would undergo exocytosis less frequently. As a direct consequence, reserves chained to reluctant vesicles would be mobilized more slowly and would behave as if constituting a slowly mobilized pool. Reserves chained to fast-releasing vesicles would be mobilized more quickly, and would be processed in parallel. In sum, the combination of two unrelated evidence-based premises of our working model predict the fully parallel arrangement outlined in *Figure 1C*.

Key for testing this idea: Distinctions between reserve pools would only emerge when the frequency of stimulation is low enough that individual reluctant vesicles remain within the readily releasable pool for extended periods of time before undergoing exocytosis. The reasoning is described in the Results section where needed to explain the design of key experiments.

Here, we begin by confirming that reserve vesicles at hippocampal synapses are segregated into multiple pools that can be distinguished by the timing of mobilization during low frequency stimulation. We then show that quickly and slowly mobilized reserves do not intermix, even when the frequency of stimulation is high, confirming that the two types are processed in parallel. The result provides a simplifying new constraint on the dynamics of vesicle recycling within presynaptic terminals.

## Results

We originally developed our working model to account for results from both primary cell culture and ex vivo slices (*Stevens and Wesseling, 1999b*; *Garcia-Perez et al., 2008*; *Gabriel et al., 2011*; *Mahfooz et al., 2016*; *Raja et al., 2019*). We chose to use cell cultures for testing the prediction that quickly and slowly mobilized reserve pools are processed in parallel because cultures are better suited for staining and destaining synaptic vesicles with FM-dyes, which can be used to distinguish between reserve pools as shown below.

In a first set of experiments, diagrammed atop *Figure 2*, we began by staining vesicles within presynaptic terminals with 60 s of 20 Hz electrical stimulation (1200 pulses) during extracellular bath application of FM4-64. We then removed the FM4-64 and washed with Advasep-7 or Captisol, which are closely related β-cyclodextrins that facilitate dye clearance from membranes (*Kay et al., 1999*).

The stain followed by wash procedure is thought to leave dye within nearly all of the vesicles that can recycle, and eliminate dye bound to non-vesicular membrane that would cause background

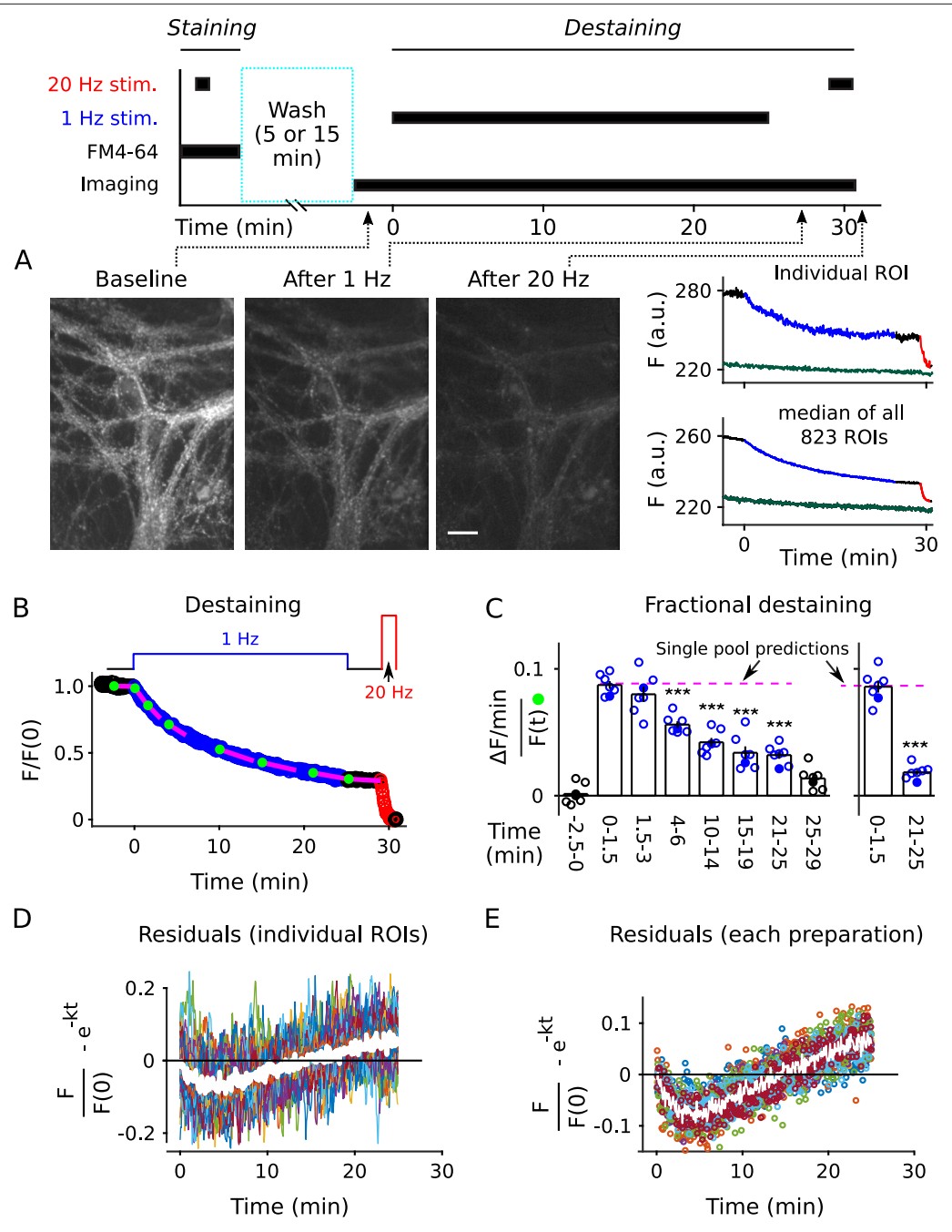

**Figure 2.** Analysis of FM4-64 destaining during 25 min of 1 Hz stimulation. (**A**) Each image is the mean of 20 sequential raw images; scale is 20 μm. The traces pertain to 1 of the 7 experiments; the lower trace (dark green) in each plot is background; *F (a.u.)* signifies arbitrary units of fluorescence. Destaining is color coded in blue for 1 Hz stimulation and red for 20 Hz here and throughout, except where indicated. (**B**) Mean ± s.e.m. of median values versus time for n = 7 preparations; error bars are smaller than the symbols. See Materials and methods for automatic detection of ROIs and formula for averaging across preparations. (**C**) Fractional destaining for a variety of time intervals; the values were calculated by dividing the slopes of the magenta lines by the values of the preceding green circles in Panel **B**. Filled symbols indicate measurements from example in Panel **A**. Rightmost two bars are after subtracting the baseline value measured either immediately before (minute -2.5 - 0) or immediately after (minute 25 - 29) 1 Hz stimulation. The dashed magenta line is the value expected from models with a single, quickly mixing reserve pool (*** is p < E - 4 compared to the first 1.5 min interval; paired t-test; *Figure 2—figure supplement 5* shows that the duration of the intervals did not affect the overall result). (**D**) Representative residual values for individual ROIs after subtracting the best fitting single exponential. For clarity, values for only 100 of 6252

*Figure 2 continued on next page*

*Figure 2 continued*

ROIs are plotted, but were chosen at random after excluding outliers with maximum deviation from zero of > 0.25 (outliers were 23% of total). Plots are moving averages of the raw residuals with a sliding window of 5. The white line is the mean of the entire data set, including outliers. (**E**) Residual values for all seven preparations.

The online version of this article includes the following figure supplement(s) for figure 2:

**Figure supplement 1.** Equivalent destaining for FM4-64 and FM2-10 in presence of Captisol.

**Figure supplement 2.** Analysis of FM4-64 signal remaining after 20 Hz stimulation for 100 s.

**Figure supplement 3.** Graphical user interface for semi-automatic ROI detection.

**Figure supplement 4.** No evidence for photobleaching.

**Figure supplement 5.** Replot of *Figure 2B–C* except with 1.5 min intervals throughout.

**Figure supplement 6.** Extensive heterogeneity among synapses.

**Figure supplement 7.** Procedure for calculating residuals after fitting with a single exponential.

**Figure supplement 8.** Double exponential fit during 1 Hz stimulation.

fluorescence (*Betz et al., 1992*; *Ryan and Smith, 1995*; *Chi et al., 2001*; *Gaffield and Betz, 2006*). The vesicles then retain the dye until undergoing exocytosis, after which they likely destain completely as the dye is washed away into the extracellular space (*Figure 2—figure supplement 1*). Because of this, the time course of bulk destaining during subsequent stimulation is likely a straightforward measure of mobilization of the vesicles that are stained and avoids complications caused by ongoing pool replenishment that are inherent to electrophysiology and synaptopHluorin imaging techniques.

To asses the timing during low frequency stimulation, we monitored destaining while stimulating at 1 Hz for 25 min (1500 pulses; *Figure 2A–B*, blue). After a 4-min rest interval (black), we completed destaining to a low level with 100 s of 20 Hz stimulation (2000 pulses, red; see also *Figure 2—figure supplement 2*). The time courses were quantified as the median fluorescence intensity of punctal regions of interest (ROIs; see *Figure 2—figure supplement 3*) after subtracting the signal remaining after the final 20 Hz train; the measurement is denoted $F/F(0)$ in *Figure 2B*. Destaining time courses were only accepted for subsequent analysis when baseline fluorescence loss was ≤ 1.5%/min, but the time courses were not corrected for the small amount that occurred in some experiments; baseline loss could have been caused by multiple factors including spontaneous exocytosis, but likely not by photobleaching (*Figure 2—figure supplement 4*).

The time courses that we measured while stimulating at 1 Hz were not compatible with single pool models or with models with multiple pools where vesicles intermix freely. That is, models where vesicles mix freely would predict that destaining at any point in time would be a constant fraction of the amount of stain still present at that time, meaning:

$$dF(t)/dt = -k \cdot F(t),$$

or

$$k = \frac{-dF(t)/dt}{F(t)}, \tag{1}$$

where $dF(t)/dt$ is the rate of change in fluorescence intensity at time $t$, $F(t)$ is the fluorescence intensity at the same time, and $k$ - the rate of *fractional destaining* - is constant over time.

The rate of fractional destaining was not constant, however, but decreased greatly over the 25 min. To quantify this, we estimated the rate at a variety of time points by dividing the rate of fluorescent decay ($dF(t)/dt$) over short intervals (slopes of the magenta lines in *Figure 2B*) by the corresponding fluorescence intensities ($F(t)$) at the beginning of the intervals (green dots). *Figure 2C*, left panel, shows that fractional destaining decreased from 0.087 ± 0.004 min$^{-1}$ (mean ± s.e.m.) during the first 1.5 min of 1 Hz stimulation to 0.032 ± 0.003 min$^{-1}$ during the interval between minutes 21 and 25, which is by a factor of 2.7 ± 0.1; that is, fractional destaining was 2.7-*fold* greater at the beginning of the 1 Hz train than at the end.

Further analysis showed that the decrease in fractional destaining was not caused by technical errors related to baseline fluorescence loss seen in some preparations, or by flaws in the premise that the final 20 Hz train fully destains all recycling vesicles. Specifically, correcting for the baseline fluorescence loss increased the estimate of the decrease in fractional destaining to a factor of 4.4 ± 0.4 (*Figure 2C*, right panel) because the correction decreased the estimate of fractional destaining for later intervals (e.g. minutes 21 - 25) by more than for earlier intervals (e.g. minutes 0 - 1.5). Likewise, correcting for any error in the premise that the final 20 Hz train eliminated dye from all recycling vesicles could increase the factor even more, although any correction of this sort would be small.

A higher resolution analysis showed that deviations from *Equation 1* occurred at almost all individual ROIs, despite variation between individuals in the details (*Figure 2—figure supplement 6*; see also *Waters and Smith, 2002*). That is, measurement noise prevented applying the analysis in *Figure 2B–C* to some of the individuals. However, *Equation 1* is mathematically equivalent to the single exponential decay:

$$F(t) = F(0) \cdot e^{-k \cdot t} \tag{2}$$

where $F(0)$ is the fluorescence intensity at the start. And, the best fitting version of *Equation 2* for each individual could be compared to the full time course of each, acquired during the full 25 min of 1 Hz stimulation; the best fitting version was obtained by allowing $k$ to vary freely between the individuals (see Materials and methods). Systematic quantification of deviations from the best fit - termed *residuals* - allowed a higher resolution analysis because the full time courses consisted of more data points than the short intervals used to calculate fractional destaining in *Figure 2C*, making them less sensitive to measurement noise (see *Figure 2—figure supplement 7* for methodology). Statistically significant deviations of p < 0.05 were detected for > 90% of individuals (5638 of 6252) in a non-parametric analysis of the residuals (*Figure 2D*); see *Figure 2E* for the analogous analysis at the level of preparations.

Even so, the fluorescence signal within some ROIs likely arose from multiple synapses. To confirm that the decrease in fractional destaining seen in populations was caused by decreases at individual synapses, we conducted additional experiments on lower density cultures where individual punctae that are clearly separated from neighbors are thought to correspond to single synapses (*Darcy et al., 2006*).

Deviations from models where vesicles mix freely were clearly apparent at many of the punctae. For example, each of the punctae indicated by magenta arrows in *Figure 3A* destained about 50% during the first 5 min of 1 Hz stimulation, but then only a small amount over the following 20 min, whereas *Equation 2* would predict nearly complete destaining. Overall, decreases in fractional destaining were measured for 83% of 235 punctae from eight preparations, which is similar to the > 90% of individual ROIs that deviated from *Equation 2* in the denser standard cultures. The amount of decrease was equivalent as well (*Figure 3B*); median fractional destaining decreased from 0.095 min⁻¹ (95% CI [0.081, 0.12]) during during the first two minutes of 1 Hz stimulation to 0.032 min⁻¹ (95% CI [0.028, 0.037]) in the interval between minutes 10 and 20 (*Figure 3C-D*), which is the same factor of 2.7 (95% CI [2.1, 3.0]) seen for standard cultures. Measurement noise contributed to uncertainty in individual fractional destaining values but would not bias the median values in either direction and, if anything, would cause an underestimate of the number of punctae where the rate of fractional destaining decreased.

~ 40% of punctae were excluded from the analysis because the linear fit of the baseline was > ±1.5%/min. The criterion was included to match the analysis of standard cultures, but disproportionately excluded individuals with low signal/noise, which might correspond to synapses that were lightly stained because of small vesicle clusters (*Schikorski and Stevens, 1997*). However: (1) equivalent decreases in fractional destaining were clearly apparent at many lightly stained punctae (*Figure 3—figure supplement 1*); (2) fractional destaining of the mean time course of the excluded punctae decreased by an equivalent amount during 1 Hz stimulation (*Figure 3C*, *yellow circles*); and (3) an analysis of residuals after fitting with *Equation 2* as in *Figure 2D–E* continued to yield statistically significant deviations in 67% of the most lightly stained quintile (i.e. 54 of the 81 least stained). Taken together, the results argue strongly against the specific concern that decreases in fractional destaining seen in the population measurements in *Figure 2* might have been caused by variation among individuals rather than decreases at individual synapses.

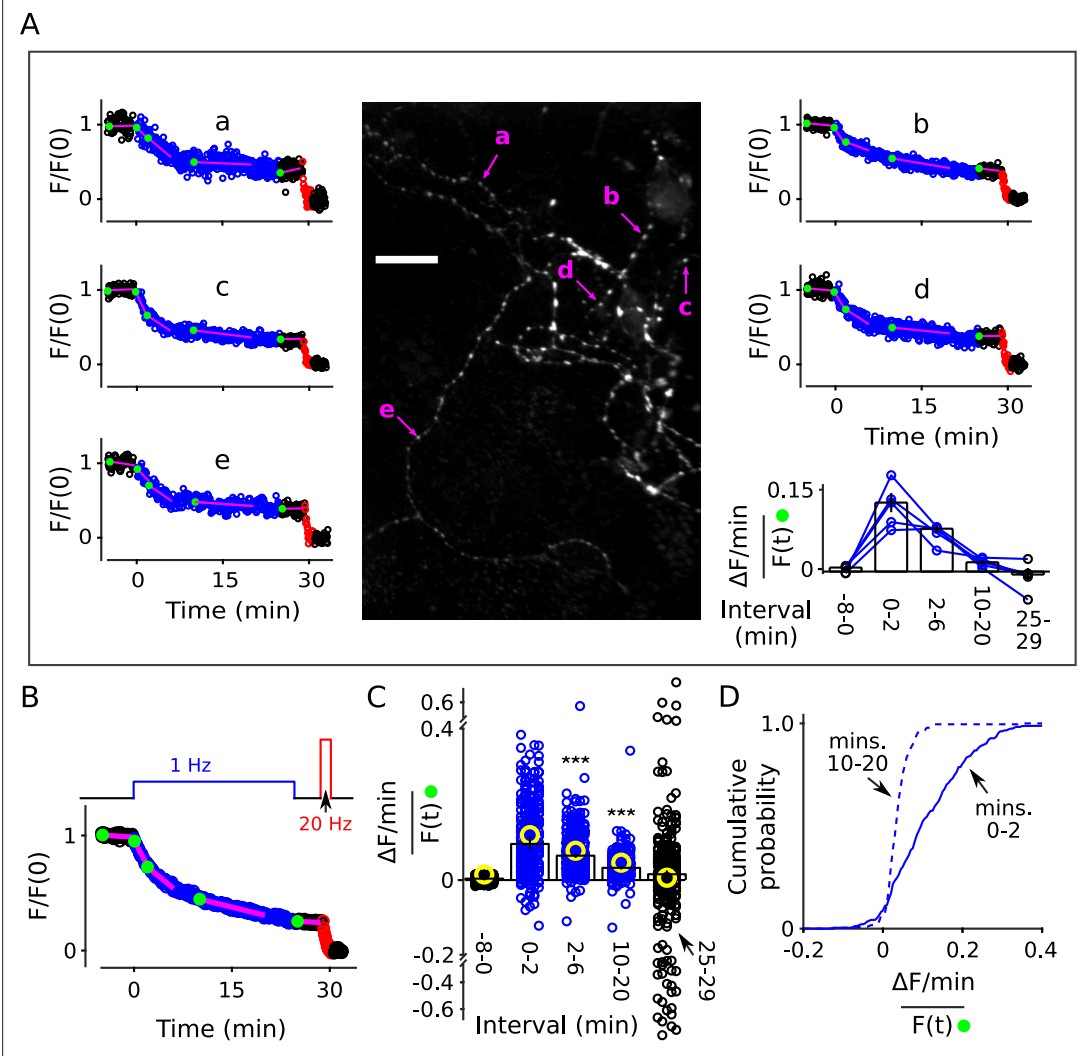

**Figure 3.** Decrease in fractional destaining during 1 Hz stimulation at individual synapses. Neurons grown at lower density were stained and destained as for *Figure 2*. For some experiments we increased the exposure time for individual images from 50 to 200 ms, which improved the signal/noise ratio, but did not result in noticeable photobleaching. Individual ROIs were limited to punctae sized less than 1.5 × 1.5 μm and clearly separate from neighbors. (**A**) Example: Image is the mean of the time-lapse during 8 min of baseline; exposures were 200 ms; scale bar is 20 μm. Plots **a**-**e** are time courses of destaining at punctae indicated by magenta arrows, after normalizing for brightness as described in Materials and methods. The magenta lines and green circles plotted on top of the time courses are analogous to *Figure 2B*, and values were used to calculate fractional destaining in the bar graph. Fractional destaining between minutes 10 and 20 was lower for the punctae in this experiment compared to the average of others reflecting extensive heterogeneity among preparations in addition to heterogeneity among punctae within individual preparations; see *Figure 2—figure supplement 6* and *Figure 3—figure supplement 2*. (**B**) Mean time course of 235 individual punctae from eight preparations where slope of baseline was ≤ 1.5%/min. (**C–D**) Fractional destaining. (**C**) Small circles (black and blue) are measurements from the 235 punctae. Yellow circles were calculated from the mean of the time courses of the 170 punctae where the slope of baseline was > 1.5% min. Scatter was greater for measurements for the 25-29 min interval because of lower signal than during the 2-6 min interval and fewer data points than during the 10-20 min interval. Bars are median ± 95% CI (*** is p < E − 9, Wilcoxon signed-rank). (**D**) Cumulative histogram of the fractional destaining values corresponding to the 0-2 and 10-20 min intervals.

The online version of this article includes the following figure supplement(s) for figure 3:

**Figure supplement 1.** Examples of destaining at lightly stained punctae.

**Figure supplement 2.** Examples showing heterogeneity among synapses.

Most individual ROIs, isolated punctae, and the collective behavior of populations could be fit with the weighted sum of two exponentials (*Figure 2—figure supplement 8*), although weighted sums of three or more exponentials and a variety of other functions could not be excluded. The result is consistent with models containing two or more reserve pools where the decreases in fractional destaining during 1 Hz stimulation would be caused by selective decrease in the number of stained vesicles in

## Box 1. Multiple reserves can explain decrease in fractional destaining.

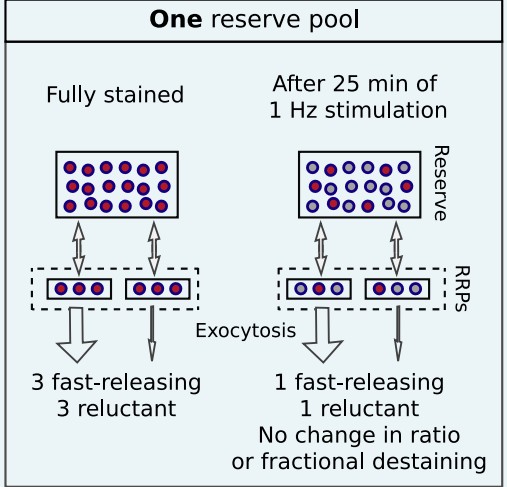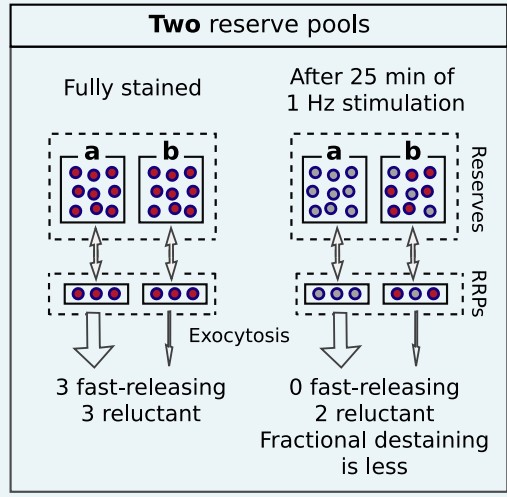

**Box 1—figure 1.** Single reserve versus two in parallel.
Comparison of schemes with one reserve pool (left) and with two parallel reserve pools that feed separate subdivisions of the readily releasable pool (right); wide arrows signify fast-releasing and narrow arrows signify reluctant readily releasable vesicles. For both schemes, the fraction of vesicles within the readily releasable pools (RRPs) and that are stained decreases over time of stimulation, accounting for the decrease in the absolute amount of destaining for each unit of time. However, with a single reserve pool, the ratio of stained fast-releasing to stained reluctant vesicles never changes, and, as a consequence, the fractional destaining at any point in time does not change. In contrast, with two reserves, vesicle mobilization in **path a** is faster than in **path b** - i.e. because the readily releasable vesicles in **path a** are fast-releasing - causing **path a** to become exhausted sooner. As a consequence, the ratio of stained fast-releasing to stained reluctant vesicles decreases over time, resulting in decreased fractional destaining.

reserve pools that are mobilized quickly, leaving the remaining dye predominantly trapped in vesicles that are mobilized slowly. *Box 1* illustrates this point with a scheme where the reserve pools operate in parallel. Additional experiments described below were required to rule out a serial organization and other explanations that do not involve multiple reserve pools at all.

### No recovery of fractional destaining during rest intervals

A key point is that mixing between the reserve pools would have to be slow during 1 Hz stimulation, if it occurs at all. Otherwise, mixing would have caused the stained vesicles in quickly and slowly mobilized reserve pools to equilibrate over the 25 min of 1 Hz stimulation, preventing the decreases in fractional destaining.

To test this, we loaded standard cultures with FM4-64 as above, then stimulated at 1 Hz for only 4 min, which was enough to decrease fractional destaining to an intermediate level (compare 2nd to 4th bars of *Figure 2C*). We then allowed the synapses to rest for 1, 3, or 8 min before resuming 1 Hz stimulation, followed by nearly complete destaining at 20 Hz, as diagrammed atop *Figure 4A*. Fractional destaining during the second 1 Hz train continued to be decreased after the rest intervals, with no indication of recovery, confirming that the decreases in fractional destaining induced by 1

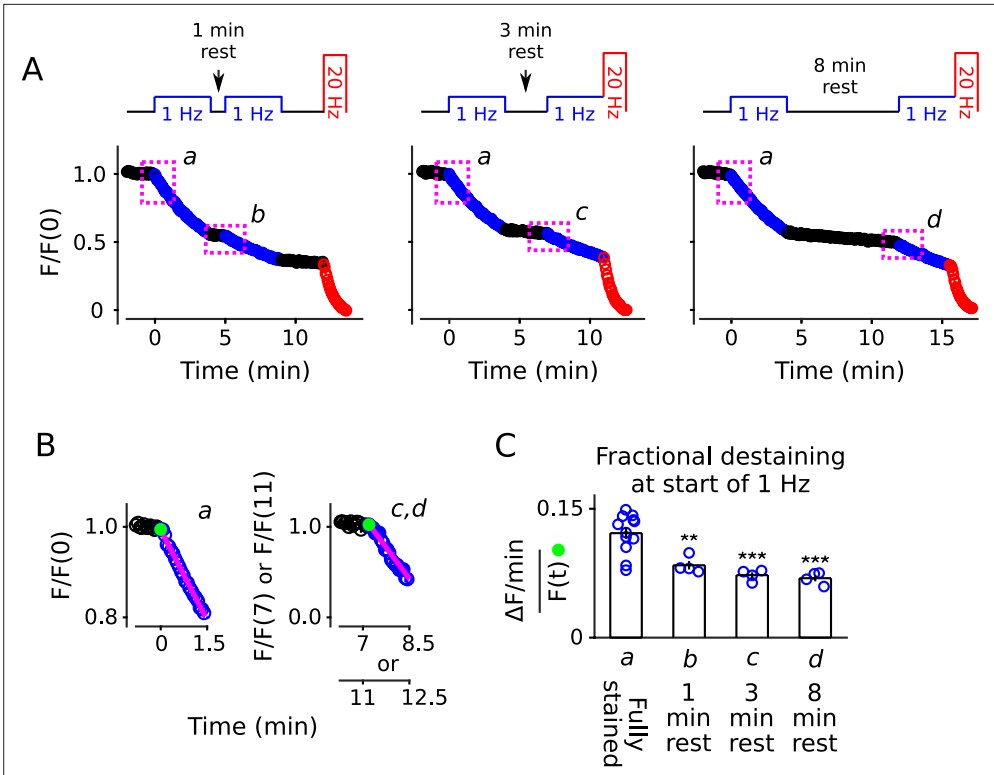

**Figure 4.** Decrease in fractional destaining induced by 1 Hz stimulation persists during long rest intervals. (**A**) Comparison of destaining during two 4 min-long trains of 1 Hz stimulation separated by 1 min, 3 min, and 8 min. (**B**) Replots of the first 1.5 min of destaining (magenta dashed boxes in Panel **A**) during 1 Hz stimulation at fully stained synapses (**a**); or after ≥ 3 min of rest following 4 min of 1 Hz stimulation (**c, d**). The fragments of the destaining time courses for **c** and **d** were renormalized by the immediately preceding rest interval to illustrate that fractional destaining was substantially less during the second 1 Hz trains. Magenta lines and green circles are slope and initial intensity as in **Figure 2B**. (**C**) Fractional destaining estimated for the first 1.5 min interval at the start of each 1 Hz train, calculated by dividing the slopes of the magenta lines by the values of the preceding green circles in Panel **B**, matching the calculation in **Figure 2C**; ** is p < 0.01 and *** is p < 0.001 compared to bar **a** (two-sample t-test).

Hz stimulation are long-lasting, at least on the time scale of minutes (**Figure 4B and C**). The result is consistent with the hypothesis tested below that vesicles are stored in multiple parallel reserve pools, but does not, by itself, rule out a wide variety of alternatives.

## Evidence against selective depletion of dye from readily releasable pools

One alternative to multiple reserve pools was the possibility that the decrease in fractional destaining during 1 Hz stimulation was caused by selective dye loss from readily releasable pools. Selective dye loss from readily releasable pools without multiple reserve pools seemed unlikely because spontaneous mixing between readily releasable and reserve pools is thought to occur on the order of one minute (**Murthy and Stevens, 1999**), which is fast compared to the destaining time course during 1 Hz stimulation (i.e. **Figure 2B**). In addition, readily releasable pools consist of an average of only 5–7 vesicles per synapse whereas the reserve pool or pools is/are thought to consist of 35–50 vesicles (**Murthy et al., 2001**; **Harata et al., 2001**). Thus, readily releasable pools would only make up 15–20% of the total recycling pool, which is substantially less than the weighting value of 0.39 for the faster of the two component exponentials in **Figure 2—figure supplement 8**.

Nevertheless, to test the possibility that the faster component of destaining was caused by selective depletion of readily releasable pools, we stained synapses as above with FM4-64, but this time destained first with a 20 Hz train of 80 pulses followed immediately by 25 min of 1 Hz stimulation

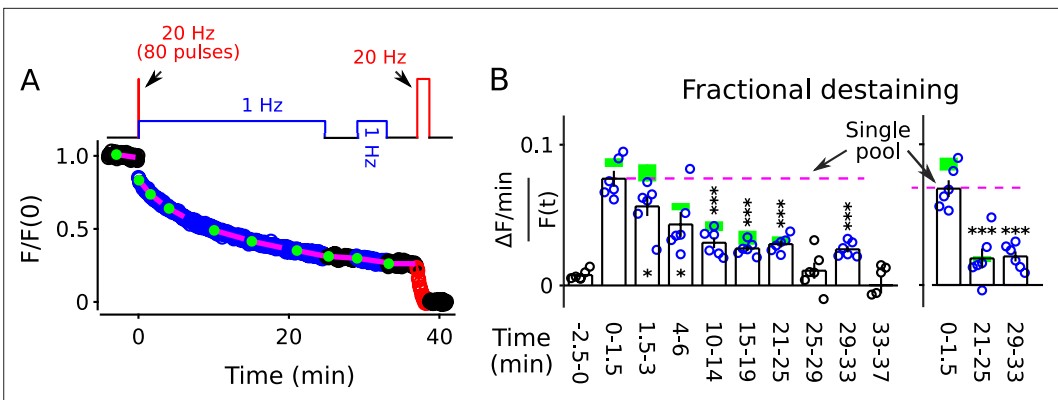

**Figure 5.** FM4-64 destaining during 25 min of 1 Hz stimulation immediately following 4 s of 20 Hz stimulation. Decrease in fractional destaining over time in Panel **B** is similar to when the initial 4 s of 20 Hz stimulation is omitted - green rectangles are mean ± s.e.m. from **Figure 2C** - ruling out selective depletion of the readily releasable pool as the cause of the decrease. Rightmost three bars are after subtracting the baseline value measured either immediately before (minute -2.5 - 0) or immediately after (minute 25 - 29) or both immediately before and after (minute 25 - 29 and minute 33 - 37) 1 Hz stimulation. The dashed magenta line is the value expected from models with a single reserve pool where fractional destaining is constant (* is p < 0.05 and *** is p < 0.001 compared to the measurement during the first 1.5 min of 1 Hz stimulation; paired t-test).

(diagram atop **Figure 5A**). The 80 pulses at 20 Hz is enough to exhaust the readily releasable pools at these synapses (**Stevens and Williams, 2007**; **Garcia-Perez et al., 2008**), but did not greatly alter fractional destaining measured during the subsequent 1 Hz stimulation (**Figure 5B**, compare blue circles to green rectangles). This result confirms that the decrease in fractional destaining seen during long trains of 1 Hz stimulation was not caused by selective depletion of dye from readily releasable pools.

For these experiments, we included a second 1 Hz train after a rest interval of 4 min (diagram atop **Figure 5A**) to test if fractional destaining would recover during rest intervals after being driven to the minimum value. No recovery was seen (compare minutes 29–33 to minutes 21–25 in **Figure 5B**, either panel). This result confirms the conclusion from **Figure 4** that decreases in fractional destaining induced by long trains of 1 Hz stimulation and measured during subsequent 1 Hz trains do not reverse quickly during rest intervals, if at all.

## Evidence against explanations that do not involve multiple reserve pools

Next we rule out alternative explanations for the long-lasting decreases in fractional destaining that do not involve pools at all.

### Long-term presynaptic depression

In principle, low-frequency stimulation can induce long-term depression in presynaptic function that might cause fractional destaining to decrease even if there were only a single reserve pool. However, no long-term presynaptic depression was seen during 1 Hz stimulation when synaptic vesicle exocytosis was measured using vGlut1-synaptopHluorin fluorescence instead of FM4-64 (**Figure 6**).

The absence of long-term depression in vGlut1-synaptopHluorin fluorescence is compatible with the decrease in fractional destaining measured with FM4-64 because, unlike FM4-64, synaptopHluorin does not disassociate from vesicle membranes after exocytosis and therefore tracks presynaptic function through multiple rounds of exo/endocytosis (**Miesenböck, 2012**). Instead, the result argues against long-term presynaptic depression as a cause for the decrease in fractional destaining measured with FM4-64. We note, however, that the result does not contradict previous reports of presynaptic long-term depression where induction required postsynaptic depolarization (**Goda and Stevens, 1996**) because postsynaptic depolarization was likely prevented in the present study by glutamate receptor antagonists.

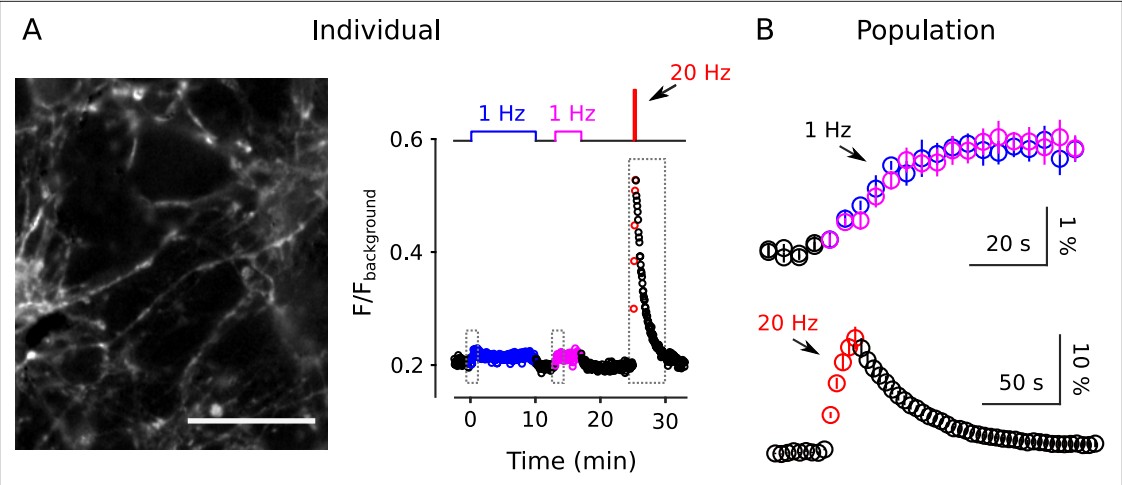

**Figure 6.** No long-term depression during 1 Hz stimulation when measured with vGlut1-SynaptopHluorin. Synapses were stimulated with two trains of 1 Hz electrical stimulation followed by 20 s of 20 Hz. (**A**) Example from a single preparation. The image is the mean of 20 sequential raw images starting with the start of 20 Hz stimulation; scale bar is 20 μm. The plot is the corresponding fluorescence signal from the entire experiment. (**B**) Mean changes in fluorescence intensity from n = 6 fields of view at the start of the two 1 Hz trains showing no difference (upper, blue is first train, magenta is second), and during the 20 Hz train (lower, red). For these experiments, fluorescence intensity was only measured over short intervals corresponding to the dashed boxes in Panel **A** - rather than during the entire experiments - to avoid photobleaching. Baseline intensity values immediately before the onset of stimulation were subtracted before combining across experiments. $\frac{\Delta F}{F}$ was 1.7 ± 0.2% at the start of the first and 1.9 ± 0.2% at the start of the second 1 Hz trains. Scale bars are $\frac{\Delta F}{F}$ versus time. The individual in Panel **A** was acquired using more light (see Materials and methods), and was part of a larger data set where a small amount of photobleaching (~ 15%) did occur over the 10 min of 1 Hz stimulation.

## Long-lasting switch to kiss and run

Alternatively, fractional destaining might decrease if endocytosis switched from the standard mode to a faster *kiss-and-run* mode, which sometimes is too fast to allow complete clearance of FM4-64 from the membrane of individual vesicles, at least in the absence of a β-cyclodextrin (***Klingauf et al., 1998***). This explanation seemed unlikely because the switch in timing of endocytosis would have to persist for minutes during rest intervals to account for the results in ***Figure 4*** and ***Figure 5***. Nevertheless, to test this, we conducted experiments similar to those documented in ***Figure 2***, except using the FM2-10 dye, which dissociates from membranes more quickly following exocytosis than FM4-64, allowing faster clearance (***Klingauf et al., 1998***).

Despite the faster clearance, the decrease in fractional destaining of the bulk signal during 1 Hz stimulation was not altered (***Figure 7***, compare blue circles to green rectangles). The result does

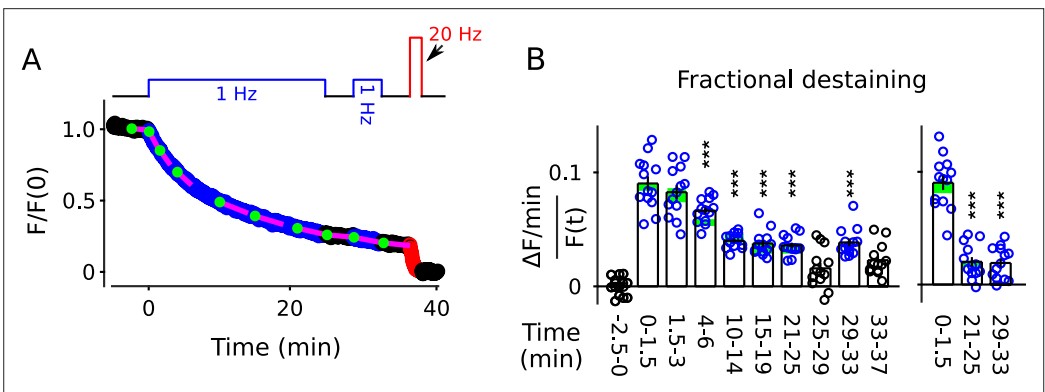

**Figure 7.** FM2-10 destaining; similar to ***Figure 2B–C*** except synapses were stained with FM2-10 instead of FM4-64 (n = 13 preparations). (**A**) Time course. (**B**) Analogous to ***Figure 2C***. Green rectangles are mean ± s.e.m. from ***Figure 2C***. Values for rightmost 3 bars were calculated by subtracting background fractional destaining values as in ***Figure 5B*** (*** is p < 0.001 compared to the measurement during the first 1.5 min of 1 Hz stimulation; paired t-test).

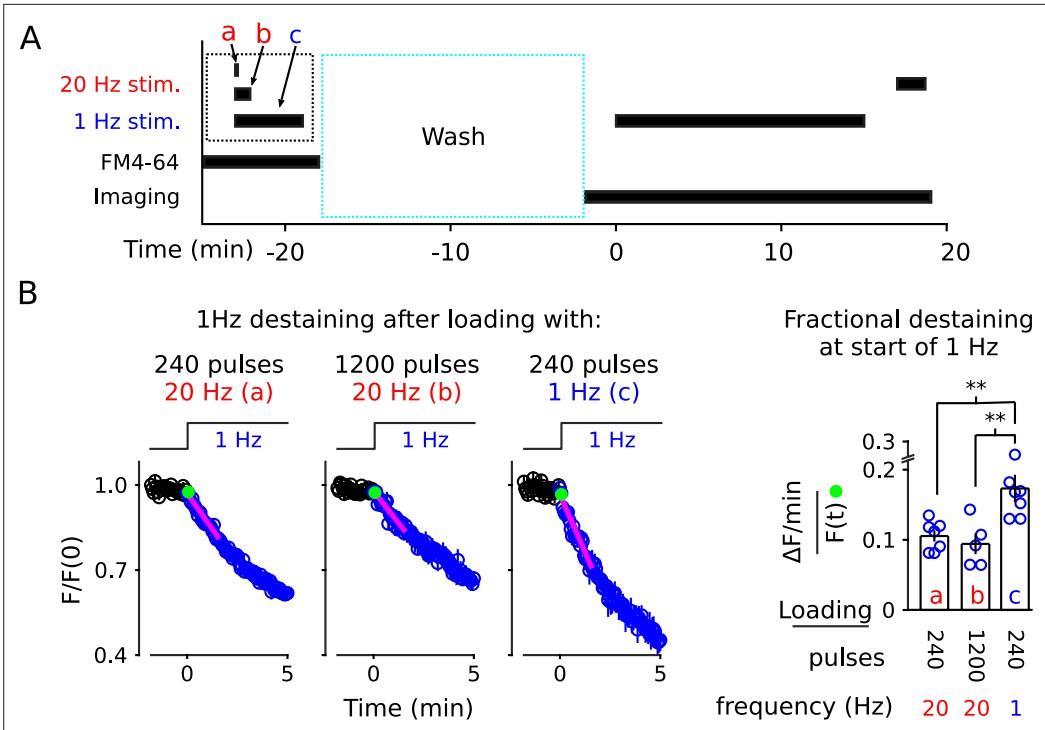

**Figure 8.** 1 Hz destaining is faster after loading at 1 Hz compared to after loading at 20 Hz. (**A**) Experimental protocol. The experimental variables are the frequency and duration of stimulation during the staining phase of the experiment (i.e., **a**, **b**, and **c**), which is different from the previous experiments. (**B**) FM4-64 destaining during the first 5 min of 1 Hz stimulation after loading with 1 Hz or 20 Hz stimulation; see *Figure 8—figure supplement 1* for full destaining time courses. Magenta lines and green circles are slope and initial intensity as in *Figure 2B* (n ≥ 5; ** is p < 0.02, two-sample t-test).

The online version of this article includes the following figure supplement(s) for figure 8:

**Figure supplement 1.** Full destaining time courses for experiments in *Figure 8*.

**Figure supplement 2.** Similar amount of staining induced by 240 pulses at 1 vs 20 Hz.

not rule out changes in the timing of endocytosis, but does argue that clearance was fast enough throughout our experiments to allow complete clearance of FM-dyes, even if changes occurred. As a consequence, the result argues against changes in the timing of endocytosis as the cause of the decrease in fractional destaining seen above. Notably, the results are not directly comparable to earlier studies conducted in the absence of β-cyclodextrins (see *Figure 2—figure supplement 1*).

The results in this section all support the hypothesis that the decrease in fractional destaining seen during long trains of 1 Hz stimulation is caused by selective depletion of a quickly mobilized pool of reserve vesicles by ruling out alternative explanations. Remaining doubt is addressed next with affirmative evidence for the existence of multiple reserve pools.

## Faster destaining when staining is induced with low-frequency stimulation

We began by reasoning that vesicles reconstituted from recycled membrane during 1 Hz stimulation would have to be predominantly targeted back to the quickly mobilized reserve pool. Otherwise, targeting to the slowly mobilized reserve pool would have displaced already stained vesicles from the slowly to the quickly mobilized reserve pool, which would have prevented the decreases in fractional destaining seen above. To test this, we stained vesicles during stimulation with 240 or 1200 pulses at 20 Hz or 240 pulses at 1 Hz in interleaved experiments. We then compared fractional destaining during subsequent 1 Hz stimulation, as diagrammed in *Figure 8A*.

Fractional destaining was greater when vesicle recycling had been driven by 1 Hz stimulation during the staining phase (compare bar **c** to **a** and **b** in *Figure 8B*), as expected if a larger fraction of

the stained vesicles were targeted to a quickly mobilized reserve pool. Follow-on experiments then showed that 240 pulses at 20 Hz and at 1 Hz stained synapses to equivalent levels (*Figure 8—figure supplement 2*), indicating that the greater fractional destaining seen after selectively staining vesicles that recycle during 1 Hz stimulation was not because fewer vesicles had been stained. These results suggest that vesicles reconstituted from membrane that is recycled during ongoing 1 Hz stimulation are targeted predominantly to a quickly mobilized reserve pool. In contrast, vesicles reconstituted during 20 Hz stimulation are targeted to both quickly and slowly mobilized reserve pools.

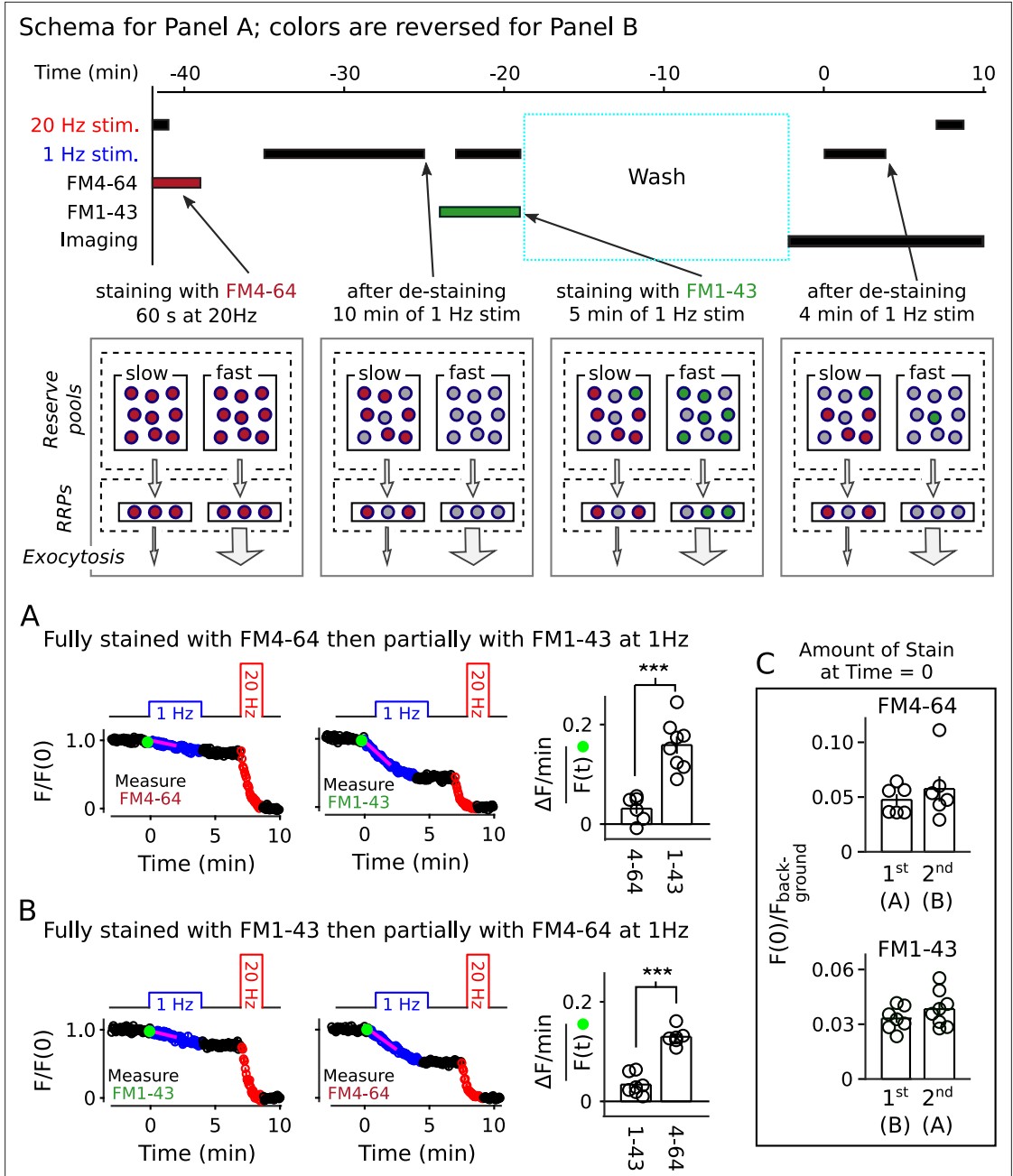

**Figure 9.** Two color separation of reserve pools. (**A**) Slowly mobilized reserve is labeled with FM4-64 (red) and quickly mobilized is labeled with FM1-43 (green); n ≥ 6; *** is p < E - 4 (two-sample t-test). (**B**) Analogous to Panel **A**, except the colors are reversed. (**A** and **B**) Magenta lines and green circles are slope and initial intensity as in *Figure 2B*. (**C**) Similar amounts of stain for each dye when applied first during 20 Hz stimulation, and then partially destained with 1 Hz stimulation, or when applied second during 1 Hz stimulation.

The online version of this article includes the following figure supplement(s) for figure 9:

**Figure supplement 1.** Description of last-in/first-out models.

## Two-color staining of separate reserve pools

The information allowed us to design a method to stain quickly and slowly mobilized reserve pools with distinct colors, and then track the mobilization separately (diagram atop *Figure 9*). To achieve this, we first stained all recycling vesicles with FM4-64 (red) using the standard 60 s of 20 Hz stimulation, then partly destained with 1 Hz stimulation for 10 min. We reasoned that 1 Hz for 10 min would nearly completely destain any quickly mobilized reserve pool because it was enough to drive fractional destaining to close to the minimum value (e.g. *Figure 2C*). We then partly re-stained the synapses with FM1-43 (green) by stimulating at 1 Hz, which would predominantly re-stain quickly mobilized reserves based on the results in *Figure 8B*. For the second staining phase, we chose a duration of 5 min, which was calculated from experiments in *Figure 2* so that the number of vesicles stained with green dye would be similar to the number, stained in red, remaining from the first staining phase. Finally, we destained first with 1 Hz stimulation to measure fractional destaining, and then with 20 Hz stimulation to fully destain both quickly and slowly mobilized reserves as usual.

As predicted, the green stain applied during 1 Hz stimulation destained > 4-fold faster than the red stain applied during the initial 20 Hz stimulation (*Figure 9A*). The results were reversed when vesicles were stained first with green at 20 Hz and second with red at 1 Hz (*Figure 9B*). Although not essential for the interpretation, *Figure 9C* confirms that the number of stained vesicles at the start of the destaining phase of the experiment was similar for the two colors.

These results show that reserve synaptic vesicles are segregated into at least two pools that can be distinguished by the timing of mobilization during 1 Hz stimulation. The one remaining caveat is that the results so far do not rule out *last-in/first-out* models where reserve vesicles are stored in a single pool with constituents that inter-mix slowly, if at all, during 1 Hz stimulation (*Rizzoli and Betz, 2004*; *Kamin et al., 2010*; see *Figure 9—figure supplement 1*). However, all serial models - including single pool last-in/first-out models - would require fast mixing either between or within reserve pools during 20 Hz stimulation to account for the results in *Figure 8B* and *Figure 9*. And, this is ruled out, below.

## Parallel mobilization

The results above did not determine whether quickly and slowly mobilized reserves are processed in series or in parallel. To distinguish between serial and parallel, we begin by showing that decreases in fractional destaining seen when stimulation is 1 Hz are absent when stimulation is 20 Hz (*Figure 10A*; compare bars **b**-**d** in *Figure 10C*). The finding is predicted by parallel models described by the scheme in *Figure 1C* - which includes our working model - because previous experiments have shown that the rate-limiting step in vesicle mobilization quickly shifts upstream from exocytosis of vesicles within the readily releasable pool to recruitment of reserve vesicles (see *Box 2*).

The result does not, by itself, rule out serial models where the absence of decreases in fractional destaining is explained by mixing mechanisms that are activated during 20 Hz stimulation (*Hilfiker et al., 1999*). However, 25 min of 1 Hz stimulation had no impact on the timing of destaining when subsequent stimulation was 20 Hz (*Figure 10B*; compare bars **f**, **g** to bars **b**, **c** in *Figure 10C*). The absence of a large impact was striking because fractional destaining would have been decreased by a factor of ~4 if measured instead during 1 Hz stimulation (e.g. *Figure 7*). This result shows that the mixing mechanisms would have to be potent enough to fully mix quickly and slowly mobilized reserves within seconds of the onset of 20 Hz stimulation. Otherwise, the time course of destaining during 20 Hz stimulation would have been shifted rightwards when initiated after 1 Hz trains that had already selectively destained the quickly mobilized reserve vesicles, which was not observed (*Figure 10D*). To our knowledge, no mixing mechanisms that operate quickly enough have been proposed.

Nevertheless, to test for a mixing mechanism in a way that avoids assumptions about timing, we first: (1) confirmed that the large decrease in fractional destaining seen during 10 min of 1 Hz stimulation persists for at least 3 min during subsequent rest intervals (*Figure 11A*, compare inset **a** vs **b**); whereas (2) equivalent destaining with 20 Hz stimulation did not alter subsequent fractional destaining (*Figure 11B*, compare inset **c** to **a** vs **b**; quantified in *Figure 11C*). We then destained synapses by stimulating: (1) at 1 Hz for 10 min to selectively destain the quickly mobilized vesicles; then (2) at 20 Hz for 12 s to activate any mixing mechanisms; and then (3) again at 1 Hz to measure any effect on fractional destaining (see diagram atop *Figure 11D*).

Fractional destaining caused by the 20 Hz stimulation was not altered by the preceding 1 Hz stimulation (*Figure 11—figure supplement 2*), confirming that the 20 Hz stimulation would have had to

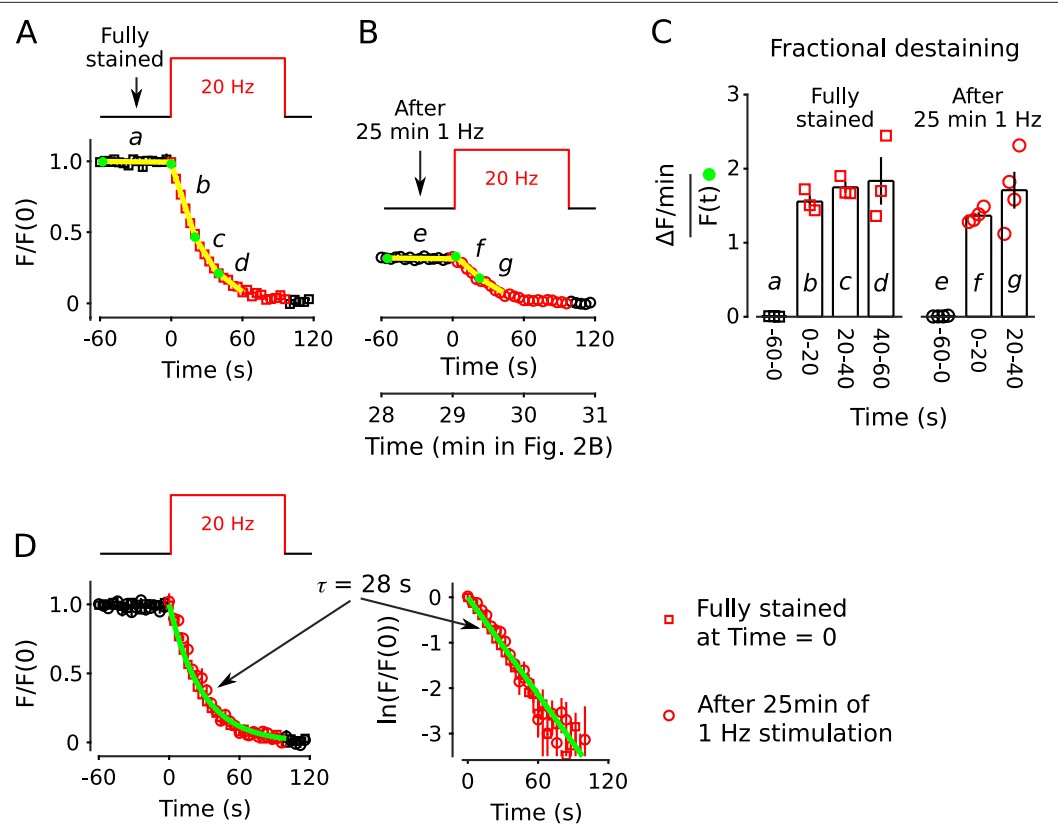

**Figure 10.** No decrease in fractional destaining when stimulation is 20 Hz. (**A**) Synapses were first stained with FM4-64 by stimulating at 20 Hz for 60 s - matching *Figure 2* and others - then destained with a single train of 20 Hz stimulation as diagrammed at top (n = 3). Yellow lines and green circles are slope and initial intensity, analogous to the magenta lines and green circles in *Figure 2B* and elsewhere. (**B**) Destaining time course after 25 min of 1 Hz stimulation; data are a subset of data plotted in *Figure 2B*; n = 4 instead of the 7 in *Figure 2* because the criterion of ≤ 1.5% destaining per min before stimulation was calculated from the rest period following the 1 Hz stimulation when the remaining stain was 3-fold less. The experiments in Panels **A** and **B** were interleaved. (**C**) Fractional destaining during 20 Hz stimulation showing no decrease over time (e.g. compare to *Figure 2C*). Note that the time intervals for calculating fractional destaining are 20 s versus ≥ 1.5 min elsewhere (i.e. because elsewhere stimulation was 1 Hz and destaining was slower). (**D**) Linear and semi-log plots of overlaid time courses after scaling to 1.0. Green lines are the single exponential described by *Equation 2* with $k = \frac{1}{\tau} = \frac{1}{28\,s}$.

The online version of this article includes the following figure supplement(s) for figure 10:

**Figure supplement 1.** Further evidence that the decrease in fractional destaining during 1 Hz stimulation is no longer evident when subsequent stimulation is 20 Hz.

have fully activated any mixing mechanism. Despite this, fractional destaining measured during the second 1 Hz train remained reduced, with no indication of any recovery (*Figure 11E*). This combination of results rules out activity-dependent mixing between quickly and slowly mobilized reserves, and, as a consequence, rules out serial models not already eliminated by the results in *Figures 2 - 10*.

In contrast, the parallel models described by *Figure 1C* and in *Box 2* do predict the absence of recovery in fractional destaining because the mechanism that causes the decrease during 1 Hz stimulation - that is, selective depletion of dye from the quickly mobilized reserve - would quickly become fully relevant again as soon as the readily releasable pool had been replenished (i.e. within tens of seconds; *Stevens and Wesseling, 1998*). Taken together, the results demonstrate that quickly and slowly mobilized reserve pools are processed in parallel.

## Modeling

Our working model, which is a specific type of parallel model, could fit the full spectrum of results in the present report using previous estimates of rate-limiting vesicle trafficking parameters (Appendix

## Box 2. Different rate-limiting mechanisms at 1 vs 20 Hz

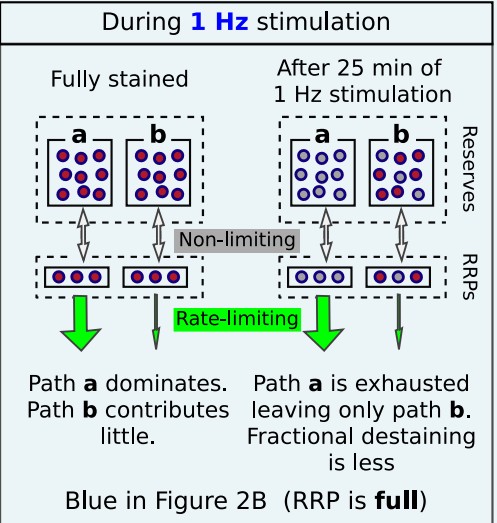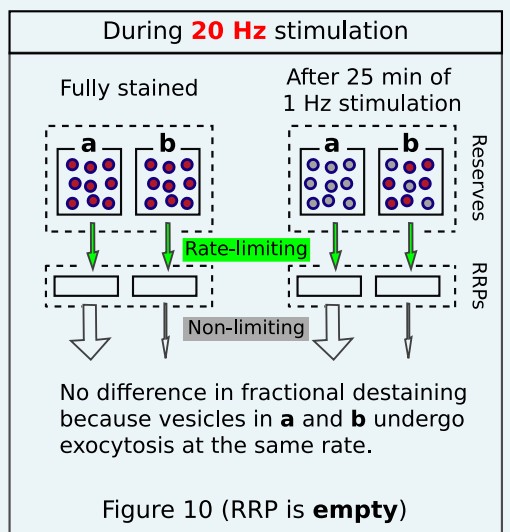

**Box 2—figure 1.** Comparison of rate-limiting mechanisms at 1 and 20 Hz.
Wide arrows signify fast-releasing and narrow arrows signify reluctant subdivisions of the readily releasable pool (RRP).The absence of rundown in fractional destaining when stimulating at 20 Hz (right panel) is in-line with parallel models because: (1) 20 Hz is intense enough to quickly drive the entire RRP - including both fast-releasing and reluctantly-releasing subdivisions - to a near-empty steady state; after which, (2) exocytosis is rate-limited by the timing of recruitment of reserve vesicles to vacant space within the subdivisions (*Wesseling and Lo, 2002*; *Garcia-Perez and Wesseling, 2008*; *Garcia-Perez et al., 2008*; *Raja et al., 2019*) instead of by mechanisms that determine probability of release of already releasable vesicles; and, finally (3) the timing of recruitment is the same for fast- and reluctantly-releasing subdivisions (*Garcia-Perez and Wesseling, 2008*; *Mahfooz et al., 2016*; see *Wesseling, 2019* for a discussion of discrepancies reported for other synapse types).

1). The analysis confirmed that parallel models can account for experimental details throughout the present study, but, given the number of parameters with otherwise unconstrained values, did not by itself provide evidence supporting our working model more than other possible parallel models. And indeed, the generic parallel model depicted in *Box 1* and *Box 2* could fit the results equally well.

### Mixing continues to be slow/absent near body temperature

The experiments above were conducted at room temperature and there is evidence that vesicles in the interiors of synaptic terminals are more motile at body temperature (*Westphal et al., 2008*; *Kamin et al., 2010*; *Lee et al., 2012*; *Park et al., 2012*). It is not known if the motility is related to rate-limiting steps in synaptic vesicle trafficking that would influence the timing of mobilization. However, the decrease in fractional destaining during 1 Hz stimulation (*Figure 12A*), and the absence of mixing during rest intervals (*Figure 12—figure supplement 1*), were both preserved at 35 C. Moreover, the time course of destaining during 25 min of 1 Hz stimulation was well fit with the weighted sum of two exponentials (*Figure 12B*), like at room temperature, and the quickly mobilized component contributed the same 39% (compare to *Figure 2—figure supplement 8*). This result suggests that the contents of quickly and slowly mobilized reserve pools are not altered by increasing the temperature. Unsurprisingly, the time constants of the two components were both less than at room temperature,

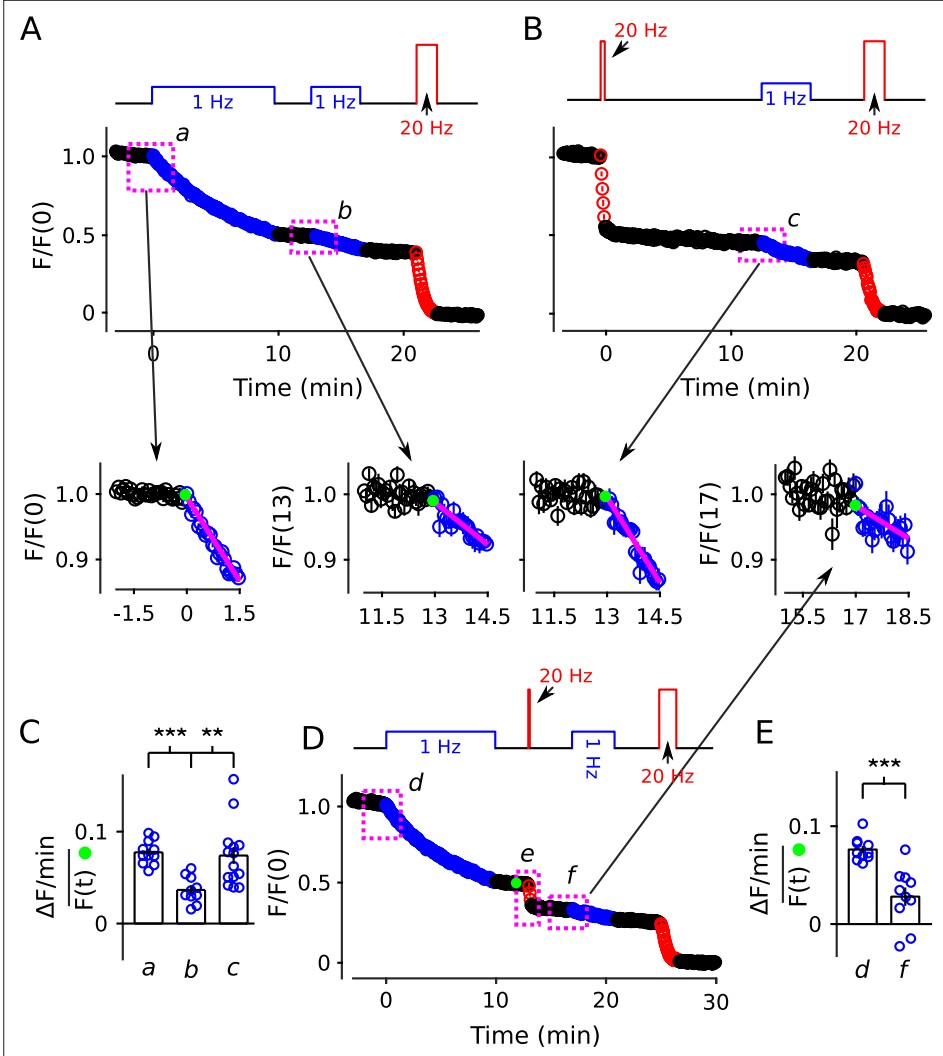

**Figure 11.** 20 Hz stimulation does not influence fractional destaining measured during 1 Hz stimulation. Synapses were first stained with FM4-64 by stimulating at 20 Hz for 60 s throughout. (**A** and **B**) Synapses were destained 50% by stimulating either at 1 Hz (Panel **A**, n = 11) or at 20 Hz (Panel **B**, n = 8), followed by a second train at 1 Hz. Insets are destaining during the first 1.5 min of the 1 Hz trains after normalizing by $F/F(0)$ during the preceding rest interval. Magenta lines and green circles are slope and initial intensity as in *Figure 2B* and elsewhere. The inset corresponding to magenta box **c** in Panel **B** summarizes results from n = 14 preparations, including the n = 8 in Panel **B** and n = 6 more from *Figure 11—figure supplement 1A* where the first 20 Hz train was delayed by 10 min. (**C**) Quantification of fractional destaining - calculated as in *Figure 2C* - during the first 1.5 min of 1 Hz stimulation in Panels **A** and **B**. The graph shows that fractional destaining during the second train was reduced when the preceding train was at 1 Hz (**a** vs **b**), but not when at 20 Hz (**a** vs **c**). The result confirms that 20 Hz stimulation depletes dye from quickly and slowly mobilized reserves with equivalent timing. ** is p < 0.01 and *** is p < E−4 (two-sample t-test). (**D**) Synapses were destained with 10 min of 1 Hz stimulation followed by 12 s of 20 Hz stimulation, then another 4 min of 1 Hz stimulation (n = 10). (**E**) Quantification of fractional destaining during the first 1.5 min of the two 1 Hz trains in Panel **D** showing that the interleaved 20 Hz stimulation did not reverse the decrease in fractional destaining induced by the first 1 Hz train. Fractional destaining in box marked **e** in Panel **D** is quantified in *Figure 11—figure supplement 2*. *** is p < 0.005 (paired t-test).

The online version of this article includes the following figure supplement(s) for figure 11:

**Figure supplement 1.** Formal control for matching *Figure 11* Panels **A** and **B**.

**Figure supplement 2.** Confirmation that fractional destaining is maximal during the interleaved 20 Hz train in in *Figure 11D*.

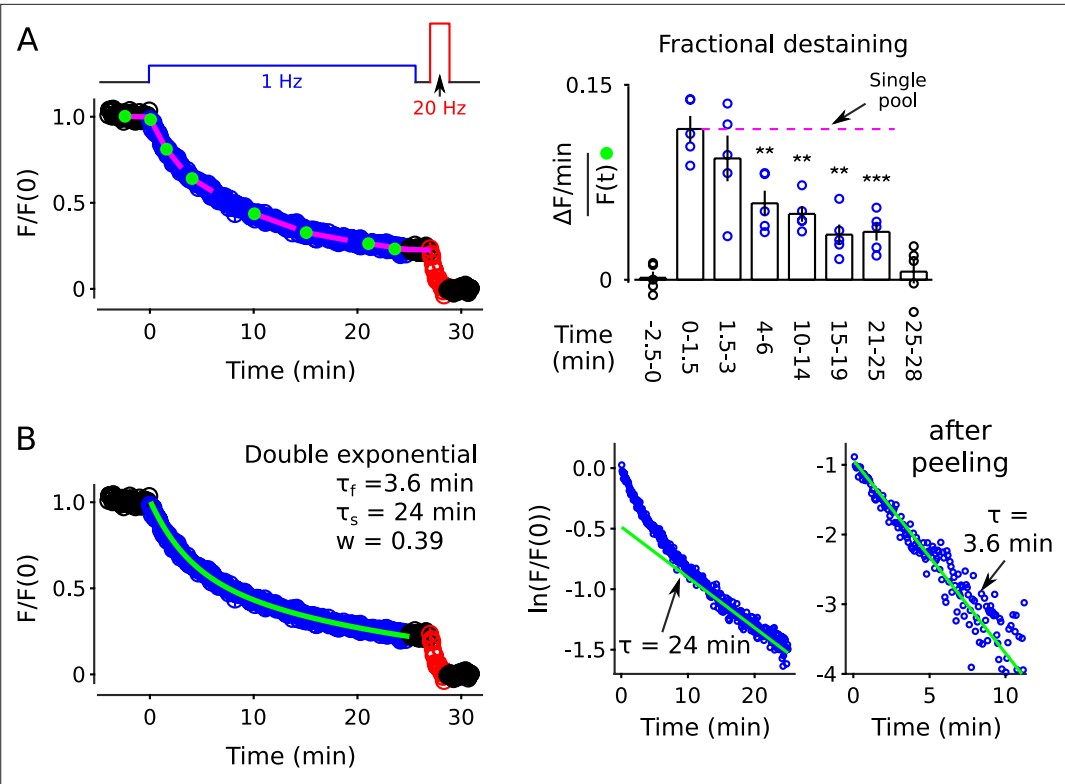

**Figure 12.** FM4-64 destaining at 35 C. (**A**) Analogous to *Figure 2B–C*, except at 35 C (n = 5, ** is p < 0.01, *** is p < 0.001, compared to the measurement during the first 1.5 min of 1 Hz stimulation, paired t-test). (**B**) Double exponential fit. Analogous to *Figure 2—figure supplement 8*.

The online version of this article includes the following figure supplement(s) for figure 12:

**Figure supplement 1.** Two color experiment at 35 C.

consistent with faster vesicular recruitment from reserve to readily releasable pools (*Garcia-Perez et al., 2008*). The temperature results are not directly relevant to whether quickly and slowly mobilized reserves are processed in series or parallel, but may be relevant to the ongoing debate about the relationship between reluctant readily releasable vesicles and asynchronous release because asynchronous release is eliminated at 35 C (*Huson et al., 2019*; see Discussion).

## Discussion

Synaptic vesicle trafficking in presynaptic terminals is central to brain function and is an increasingly important target of anti-epilepsy medicines (*Lyseng-Williamson, 2011*). A more detailed understanding might generate insight into the first principles underlying biological computation, and might aid second generation drug discovery (*García-Pérez et al., 2015*). Here, we show that quickly and slowly mobilized reserve pools of vesicles are present in hippocampal synapses, and that the two types are processed in parallel, rather than in series as previously assumed.

The experiments were designed to test predictions that distinguish our working model from current concepts (*Figure 1D*). The model emerged from two separate lines of evidence. The first line argued against the concept that mass action of reserve vesicles influences the timing of vesicle recruitment to the readily releasable pool (*Stevens and Wesseling, 1999b*; *Garcia-Perez et al., 2008*; see *Miki et al., 2020* for a recent complementary type of evidence), while simultaneously supporting the often-linked concept that depletion of reserve pools is one of the mechanisms that causes short-term synaptic depression during extensive stimulation (*Gabriel et al., 2011*). The second line supported the concept that fast- and reluctantly-releasing readily releasable vesicles differ because they are docked to distinct types of release sites rather than because they are at distinct stages of biochemical priming (*Wesseling and Lo, 2002*; *Garcia-Perez and Wesseling, 2008*; *Mahfooz et al., 2016*; *Raja*

*et al., 2019*; see also *Hu et al., 2013*; *Böhme et al., 2016*; *Akbergenova et al., 2018*; *Maschi and Klyachko, 2020*; *Li et al., 2021*; *Karlocai et al., 2021*; *Gou et al., 2022*).

The results of the current study do provide new support for our working model because: (1) the opposite conclusion - that reserves are processed in series as widely assumed - would have caused us to discard the model; and (2) no competing parallel model has yet been proposed. We emphasize, however, that the main conclusion that quickly and slowly mobilized reserve vesicles are processed in parallel does not depend on either of the lines of evidence that motivated our working model, or on the model itself, and the evidence against serial models would continue to be equally strong even if doubts arose about the previous conclusions.

## Absence of mixing

Notably, our working model contains a slow undocking mechanism operating continuously at ~1/min that would have been consistent with slow mixing between quickly and slowly mobilized reserves if the undocked chains of vesicles were to mix freely (*Gabriel et al., 2011*). Because of this, the model did not anticipate the complete absence of recovery in fractional destaining over 8 min-long rest intervals in *Figure 4*. However, the absence of mixing does not argue against the model (see Appendix 1) because chains of vesicles might be attached to a stable cytoskeletal scaffold that would be preserved after undocking, or might reside within a liquid phase gel that prevents intermixing (e.g. *Siksou et al., 2007*; *Fernández-Busnadiego et al., 2010*; *Cole et al., 2016*; *Milovanovic et al., 2018*).

## Relationship to a deep reserve

The present results do not rule out the possibility that other types of synapses additionally or instead harbor a variety of reserve pools that are connected in series (*Neves and Lagnado, 1999*; *Richards et al., 2003*). And indeed, the present results only pertain to vesicles mobilized during 60 s of 20 Hz stimulation, whereas many vesicles within hippocampal presynaptic terminals are mobilized much more slowly, if at all during this time (*Harata et al., 2001*). We refer to the vesicles that are not mobilized in the time frame of minutes as the *deep reserve*. One possibility is that deep reserve vesicles do mix slowly with one or both of the quickly and slowly mobilized reserves identified above (*Denker et al., 2011*). This might account for the timing information in *Rey et al., 2015* where vesicles that were recycled more recently were mobilized more quickly over 10's of minutes. However, very slow mixing taking 10's of minutes between the quickly and slowly mobilize reserve pools identified above cannot be ruled out either.

## Relation to multiple classes of vesicles

The experiments in the present study were designed to measure vesicle exocytosis involved in action potential evoked release, whereas transmitter released spontaneously is thought to be stored in a different class of vesicles (*Kavalali, 2015*).

The present results are consistent with the possibility that the decision about whether a recycling vesicle will become reluctant or fast-releasing upon entering the readily releasable pool is made earlier, at the time of entry into the reserve pool. Alternatively, the parallel mobilization of slow and fast reserves might be analogous to current ideas about parallel cycling of vesicles involved in spontaneous release, which would then require multiple classes of vesicles (*Raingo et al., 2012*). However, adding the concept of multiple classes of vesicles could not replace the requirement for multiple classes of release sites because, otherwise, the readily releasable pool would eventually fill with reluctantly releasable vesicles during low frequency stimulation, causing long-lasting depression, which was not seen (*Figure 6*); long-lasting depression is ruled out in a second way in *Figure 8* and *Figure 9* where destaining was faster - not slower - when the staining phase of the experiments included long trains of 1 Hz stimulation.

## Multiplexed frequency filtering

Finally, our working model explains slowly mobilized reserves as a logical consequence of multiple classes of release sites, with no functional significance of their own. However, we anticipate that the variation among synapses in *Figure 2—figure supplement 6* and *Figure 3—figure supplement 2* will nevertheless be relevant for understanding biological computation.

That is, inefficient release sites that engender reluctantly releasable components of the readily releasable pool would function as high-pass/low-cut frequency filters when transmitting information encoded within presynaptic spike trains, whereas efficient release sites engendering fast-releasing components would function as low-pass/high-cut filters (*Mahfooz et al., 2016*). The presence of multiple types of release sites might therefore endow individual synapses with a mechanism for selectively transmitting multiple types of frequency information, analogous to *multiplexing* in digital information technology. The variation among synapses in the time course of changes in fractional destaining observed here during low frequency stimulation suggests that individual synapses in vivo likely contain machinery for modulating multiplexing. If so, this machinery would have the capacity to store substantially more information than mechanisms that always affect synaptic connection strength evident during low frequency use (*Bartol et al., 2015*). Notably, we previously observed extensive variation between calyx of Held synapses in the ratio of fast-releasing to reluctant readily releasable vesicles, which, when taken together with the present results, suggests that mechanisms for modulating multiplexing might be available at a wide range of synapse types (*Mahfooz et al., 2016*).

### Relation to asynchronous release

The multiplexed frequency filtering hypothesis depends critically on the premise that exocytosis of reluctantly releasable vesicles is tightly synchronized to action potentials, and at least one study concluded that this is not the case at developing calyx of Held synapses (*Sakaba and Neher, 2001*). However, our own experiments indicated that reluctant vesicle exocytosis at calyx of Held is tightly synchronized to action potentials at later ages (*Mahfooz et al., 2016*). And, raising the temperature to 35°C in the hippocampal cell culture preparation used here eliminates *asynchronous release* (*Huson et al., 2019*), but does not alter the size or decrease mobilization of the slowly mobilized reserve pool (compare *Figure 12B* to *Figure 2—figure supplement 8*).

## Materials and methods

Culture and imaging methods were similar to *Chowdhury et al., 2013* (cultures) and *Raja et al., 2019* (imaging).

### Cultures

All procedures were conducted in accordance with the European and Spanish regulations (2010/63/UE; RD 53/2013) and were approved by the Ethical Committee of the Generalitat Valenciana (2022/VSC/PEA/0255). Neurons were dissociated from hippocampi of 0- to 2-day-old mice of both sexes, and grown on class coverslips coated with a mixture of laminin and polyornithine.

### Imaging

Imaging was performed 11–21 days after plating, at 0.25 Hz, using a CCD camera (Photometrics CoolSnap HQ), and 25X oil objective (Zeiss 440842–9870), except where indicated. Focal drift was avoided in most experiments by clamping focal distance by feeding back the signal from a distance sensor attached to the objective (Physik Instrumente D-510.100, ~ 1 nm resolution) to a piezoelectric objective drive (MIPOS250, PiezosystemJena). For most experiments, FM4-64 was imaged with a green LED (530 nm; 50 ms exposures for most experiments; 200 ms exposures for 3 of 8 experiments for *Figure 3*) via the XF102-2 filter set from Omega. When used in combination with FM1-43, FM4-64 was instead imaged with an amber LED (590 nm, 1 s exposures) via a custom set containing band pass excitation filter FB600-10, dichroic DMLP638R, and long-pass emission filter FELH0650, all from Thorlabs. FM1-43 was imaged with a blue LED (470 nm; 200 ms exposures) via the XF100-2 filter set from Omega and FM2-10 was imaged with the same blue LED (200 ms exposures), but with the XF115-2 filter set from Omega. vGlut1-synaptopHluorin (*Voglmaier et al., 2006*) was expressed by infecting at day 7 after plating with an AAV1 construct and imaged with the blue LED via the XF116-2 filter set from Omega; exposure length was 200 ms except for *Figure 6A* where exposure length was 300 ms and the objective was 60X (Olympus UPlanFL N) rather than 25X. LEDs were all Luxeon Star and were driven with 700 mA of current. Bathing solution was exchanged continuously during imaging at 0.2-0.5 ml/min within a sealed chamber holding ~ 35 µL. Heating to 35°C was monitored with a bead

thermistor (Warner, TA-29) built into the chamber, and exposed to the extracellular solution. Electrical stimulation was bipolar (0.5 ms at - 30 V then 0.5 ms at + 30 V) via two platinum electrodes built into the chamber. Neurotransmitter receptors were blocked with (in µM): picrotoxin (50); DNQX (10); and DL-APV (50). Other solutes were (in mM): NaCl (118); KCl (2); $CaCl_2$ (2.6); $MgCl_2$ (1.3); Glucose (30); and HEPES (25). FM4-64 and FM1-43 were used at 15 µM, and FM2-10 at 100 µM. Advasep-7 and Captisol were purchased from Cydex Pharmaceuticals or graciously provided as samples and used at 1 mM during the destaining phase of FM-dye experiments and throughout experiments using vGlut1-synaptopHluorin.

## Processing

Time lapse images were aligned and Regions of Interest (ROIs) identified automatically as described in *Raja et al., 2019* (see *Figure 2—figure supplement 3*), except for *Figure 3* where they were detected manually. Between 175 and 1773 ROIs were detected for each field of view (except for *Figure 3*). For summary statistics, the median values from a single field of view/experiment were counted as n = 1 unless otherwise indicated.

## Normalization

For comparing images across preparations - or between individual punctae for *Figure 3* - median or individual ROI values were: divided by the mean value of the background region and then normalized by the baseline signal.

## Curve fitting

The Matlab code for identifying the best fitting single exponential was:

```
function k=GetBestSingleExp(DcyDat, dpmin)
    s=fitoptions('Method','NonLinearLeastSquares',…
                'Lower', 0, 'Upper', 0.3, 'Startpoint', -0.002);
    f=fittype('exp((k*-1)*t)','options',s, 'independent', 't');
    Time = (0:(length(DcyDat)-1))/dpmin;
    cfun = fit(Time',DcyDat,f);
    k = cfun.k;
```

## Acknowledgements

We thank Dr. Silvio Rizzoli for help understanding relationships between previous models of synaptic vesicle cycling, and Drs. Artur Llobet, Francisco Martini, Donald Lo, William Wetsel, Robert Renden, Jay Coggan, and Ana Gomis for advice about the writing. This work was funded by: the Generalitat Valenciana Prometeo Excellence Program (CIPROM2022/8); the Ministry of Science of Spain (SAF2013-48983R, BFU2016-80918-R, and PID2019-111131GB-I00); and the Unión Temporal de Empresas (UTE) project at the Centro de Investigación Médica Aplicada of the Universidad de Navarra. The funders had no role in study design, data collection and analysis, or preparation of the manuscript.

## Additional information

### Funding

| Funder | Grant reference number | Author |
| --- | --- | --- |
| Generalitat Valenciana | CIPROM2022/8 | Isabel Perez-Otano John F Wesseling |
| Agencia Estatal de Investigación | PID2019-111131GB-100 | John F Wesseling |
| Agencia Estatal de Investigación | BFU2016-80918-R | John F Wesseling |

| Funder | Grant reference number | Author |
|---|---|---|
| Agencia Estatal de Investigación | SAF2013-48983R | Isabel Perez-Otano<br>John F Wesseling |

The funders had no role in study design, data collection and interpretation, or the decision to submit the work for publication.

## Author contributions

Juan Jose Rodriguez Gotor, Formal analysis, Investigation, Methodology, Writing – review and editing; Kashif Mahfooz, Formal analysis, Validation, Investigation, Methodology, Writing – review and editing; Isabel Perez-Otano, Conceptualization, Funding acquisition, Writing – original draft, Writing – review and editing; John F Wesseling, Conceptualization, Resources, Data curation, Software, Formal analysis, Supervision, Funding acquisition, Validation, Investigation, Visualization, Methodology, Writing – original draft, Project administration, Writing – review and editing

## Author ORCIDs

Juan Jose Rodriguez Gotor http://orcid.org/0000-0002-3838-1115
Kashif Mahfooz http://orcid.org/0000-0002-9218-6927
John F Wesseling http://orcid.org/0000-0002-7565-2594

## Ethics

All procedures were conducted in accordance with the European and Spanish regulations (2010/63/ UE; RD 53/2013) and were approved by the Ethical Committee of the Generalitat Valenciana (2022/ VSC/PEA/0255).

Joint Public Review: https://doi.org/10.7554/eLife.88212.3.sa1
Author response https://doi.org/10.7554/eLife.88212.3.sa2

# Additional files

## Supplementary files

• MDAR checklist

## Data availability

Source data can be downloaded from https://doi.org/10.5061/dryad.zcrjdfn8s.

The following dataset was generated:

| Author(s) | Year | Dataset title | Dataset URL | Database and Identifier |
|---|---|---|---|---|
| Wesseling J | 2024 | Data for: Parallel processing of quickly and slowly mobilized reserve vesicles in hippocampal synapses | https://doi.org/10.5061/dryad.zcrjdfn8s | Dryad Digital Repository, 10.5061/dryad.zcrjdfn8s |

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

# Appendix 1

## Destaining time courses fit with working model in *Figure 1D*

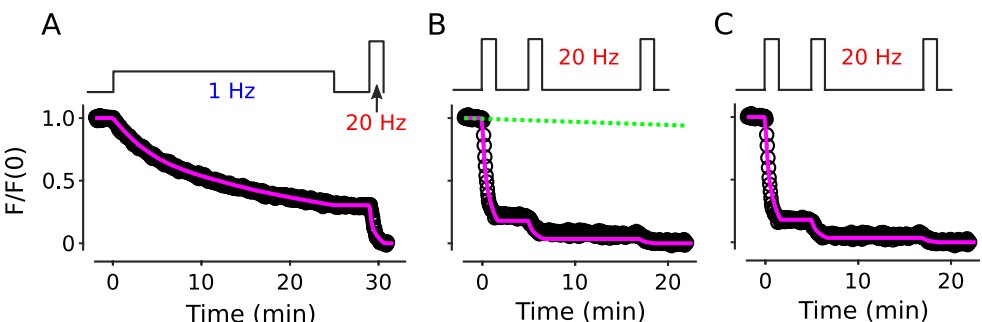

**Appendix 1—figure 1.** Magenta lines are the simulation with three types of release sites with $p_{rs}$ = 0.28 (7%), 0.035 (33%), and 0.0025 (60%). (**A**) Replot of results in *Figure 2B* confirming that the model is compatible with FM-dye destaining when stimulation is 1 Hz. (**B**) Replot of results in the left panel of *Figure 2—figure supplement 2A* after re-normalizing so that the final points have a value of 0. The near-miss of the magenta line illustrates how highly constrained the model is for destaining during 20 Hz stimulation since there are essentially no free parameters. The green line is the estimated stimulation-independent rundown of 0.25 %/min. (**C**) Replot after correcting for the rundown, confirming that the model is compatible with FM-dye destaining when stimulation is 20 Hz.

Parameter values specified by the results from the original electrophysiological experiments - see *Gabriel et al., 2011* - were the length of docked and undocked *tethering units* (4 vesicles, including the readily releasable vesicle when docked), the timing of recruitment of a vesicle from a docked tether unit to the release site when vacant (0.13 s⁻¹ at rest, accelerating to 0.22 s⁻¹ during 20 Hz stimulation - alpha in the Matlab code below), and the timing with which tethering units are replaced (0.017 s⁻¹ at rest, accelerating to 0.025 s⁻¹ during 20 Hz stimulation - gamma in Matlab code). However, the original results only specified parameters that are rate-limiting for neurotransmitter release during intense stimulation, and there were a variety of additional parameters - relevant to vesicle trafficking, but not rate-limiting for release - that were relevant to FM-dye destaining, especially during low frequency stimulation.

The most relevant additional parameters pertained to the release sites and included: (1) the number of types of release sites; (2) the probability with which each type catalyzes exocytosis after single action potentials (i.e., $p_{rs}$ for *probability of release* for the *release site* when occupied by a readily releasable vesicle, which is equivalent to $p_{v,hi}$ and $p_{v,lo}$ in *Mahfooz et al., 2016*, except with the possibility for more than two types of release sites); and, (3) the relative amount of each type.

In addition, we fixed the number of non-docked tethering units that can exchange with each docked unit at 1, meaning that each release site would process vesicles from two tethering units (8 vesicles). The value was chosen because depletion of the readily releasable pool destained synapses by slightly more than 1/8 (0.15 ± 0.01 in *Figure 5*). This implies that a typical synapse with 7 release sites would contain 7X 8 = 56 recycling vesicles, which is in-line with results in *Ryan et al., 1997*; *Harata et al., 2001*; *Schikorski and Stevens, 2001*.

Next, for fitting destaining during 20 Hz stimulation, it was necessary to allow > 5-fold facilitation, at least for the release sites with the lowest values for $p_{rs}$, because otherwise 20 Hz stimulation would not exhaust the readily releasable pool: exhaustion during 20 Hz stimulation is verified experimentally in Figure 2 of *Garcia-Perez et al., 2008*; and > 5-fold facilitation is verified in Figure 2 of *Stevens and Wesseling, 1999a*. We had facilitation increase with a single exponential with rate parameter of 30 action potentials to maintain consistency with Figure 2 of *Stevens and Wesseling, 1999a*, but the precise timing was not a key factor because of the flexibility provided by the three release site parameters.

Finally, we reasoned that previously docked tethering units would need to have the capacity to re-dock to the same release site. Otherwise: Either a large number of dye-stained vesicles would be trapped within synaptic terminals during long trains of low frequency stimulation; or quickly and slowly mobilized reserves would mix on the time course of minutes. A large number of trapped

vesicles is ruled out in *Figure 2A* and elsewhere, and mixing is ruled out in *Figure 4* and *Figure 9*, and elsewhere.

FM-dye destaining curves (*Appendix 1—figure 1*) could be well fit by including as few as two types of release sites, but at least three types were required to match additionally the short-term depression seen in synaptic strength during 20 Hz stimulation (*Appendix 1—figure 2*).

The Matlab code for simulating the model for a single release site was:

```
function [DestainTimeCourse, SynapticStrengthTimeCourse] = ...
DestainReleaseSite(NumberOfTetherUnits, Time_seconds, ...
                   NumberVesiclesPerUnit, prs_VsTime, ...
                   alpha_VsTime, gamma_VsTime, ...
                   ActionPotentials_VsTime)
    DestainTimeCourse(length(Time_seconds)) = nan;
    SynapticStrengthTimeCourse(length(Time_seconds)) = 0;
    TetherUnits(1:NumberOfTetherUnits, 1:NumberVesiclesPerUnit) = 1;
         %All vesicles in all tether units are stained to start
         %Each space on tether unit can be:
         %                                    1 (full, stained)
         %                                   -1(full, unstained)
         %                                    0 (vacant)
    DeltaTime_seconds = Time_seconds(2)-Time_seconds(1);
    DockedUnit = 1;
    for i = 1:length(Time_seconds)
        DestainTimeCourse(i) = sum(TetherUnits(:)==1);
        %fluorescence is equivalent to
        %the number of vesicles that are stained
        if (TransitionP(gamma_VsTime(i), DeltaTime_seconds))
            [DockedUnit, TetherUnits] =
                SwitchDockedUnit(DockedUnit, ...
                          1:NumberOfTetherUnits, ...
                          TetherUnits);
        end
        if (TetherUnits(DockedUnit,end)==0)
            if (TransitionP(alpha_VsTime(i),DeltaTime_seconds))
                TetherUnits = AdvanceTetherUnit(DockedUnit, TetherUnits);
        end
    end
    if (ActionPotentials_VsTime(i) && ...
        (TetherUnits(DockedUnit,end)~=0) && ...
        (prs_VsTime(i)) > rand())
            TetherUnits(DockedUnit,end) = 0;
            SynapticStrengthTimeCourse(i) = 1;
    end
 end

function P = TransitionP(DeltaTime_seconds, rateconstant)
    P = ((1-exp(-rateconstant*DeltaTime_seconds)) > rand());

function [NewDocked, TetherUnits] = ...
            SwitchDockedUnit(CurrentlyDocked, AllChoices, TetherUnits)
    AllChoices(AllChoices==CurrentlyDocked) = [];
    NewDocked = randsample(AllChoices, 1);
    TetherUnits(NewDocked, TetherUnits(NewDocked, :)==0) = -1;
    % The preceding line replaces vacant spaces
    % on tethers with unstained vesicles
```

```
function TetherUnits = AdvanceTetherUnit(DockedUnit, TetherUnits)
   if (TetherUnits(DockedUnit,end) ~= 0) %Only advance if vacant
      warning('Trying to advance a chain with no vacancy at end');
   else
      for i = length(TetherUnits(DockedUnit,:)):-1:2
         TetherUnits(DockedUnit, i) = TetherUnits(DockedUnit,i-1);
      end
      TetherUnits(DockedUnit, 1) = 0; %This makes the end vacant
 end
```

Simulations were repeated 1000 times for each type of release site for *Appendix 1—figure 1* and 100,000 times for *Appendix 1—figure 2* before averaging and then calculating the weighted sum across types of release sites.

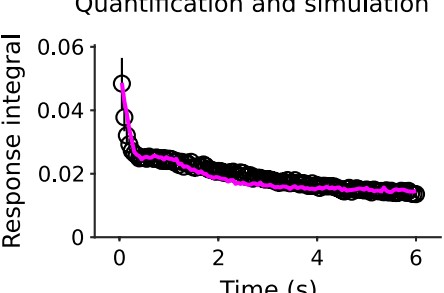

**Appendix 1—figure 2.** Electrophysiological recordings of synaptic responses between pairs of neurons in culture during 6 s of 20 Hz stimulation, and corresponding simulation using the model in *Appendix 1—figure 1* (magenta line). The electrophysiological trace is the average across 11 pairs, including the pairs that constituted the untreated wildtype control in Figure 5 of *García-Pérez et al., 2015*. The inset shows the first 4 responses on an expanded time scale. Outer scale bars pertain to the entire trace and are 500 pA by 1s; inner bars pertain to the inset and are 500 pA by 50 ms. Responses were quantified by integrating each interval after subtracting the baseline before the first response (dashed cyan line), so that later responses included a substantial component caused by asynchronous release (*Hagler and Goda, 2001*). Results for each cell pair were normalized by the sum of the first 40 responses (2 s) before combining across cell pairs; we normalized this way because of extreme variation between cell pairs in the short-term plasticity seen during the first few responses - the paired pulse ratio of the first two responses varied from 1.27 to 0.64 - and because 40 responses has been used elsewhere to approximate readily-releasable pool size (*Murthy and Stevens, 1999*). The 80 responses to trains designed to ensure exhaustion above would be an overestimate because of ongoing recruitment to vacant release sites (*Wesseling and Lo, 2002*).

