## [Editor Report · eLife assessment]

This study addresses the long-standing question as to how different functional pools of synaptic vesicles are organized in presynaptic terminals to mediate different modes of neurotransmitter release. Based on imaging of active synapses with recycling synaptic vesicles labeled by FM-styryl dyes, the authors provide data that are compatible with the hypothesis that two separate reserve pools of vesicles – slowly vs. rapidly mobilizing – feed two distinct releasable pools – reluctantly vs. rapidly releasing. Overall, this study represents a **valuable** contribution to the field of synapse biology, specifically to presynaptic dynamics and plasticity. The authors' methodological approach of using bulk FM-styryl dye destaining as a readout of precise vesicle arrangements and pools in a population of functionally very diverse synapses has limitations. Consequently, the evidence that directly supports the authors' two-pool-interpretation of their data is **incomplete**, and alternative interpretations of the data remain possible.

---

## [Referee Report · Joint Public Review]

This study is concerned with the general question as to how pools of synaptic vesicles are organized in presynaptic terminals to support different types of transmitter release, such as fast synchronous and asynchronous release. To address this issue, the authors employed the classical method of loading synaptic vesicle membranes with FM-styryl dyes and assessing dye destaining during repetitive synapse stimulation by live imaging as a readout of the mobilization of vesicles for fusion. Among other findings, the authors provide evidence indicating that there are multiple reserve vesicle pools, that quickly and slowly mobilized reserves do not mix, and that vesicle fusion does not follow a mono-exponential time course, leading to the notion that two separate reserve pools of vesicles - slowly vs. rapidly mobilizing - feed two distinct releasable pools - reluctantly vs. rapidly releasing. These findings are valuable to the field of synapse biology, where the organization of synaptic vesicle pools that support synaptic transmission in different temporal and stimulation regimes has been a focus of intense experimentation and discussion for more than two decades.

On the other hand, the present study has limitations, so that the authors' key conclusions remain incompletely supported by the data, and alternative interpretations of the data remain possible. The approach of using bulk FM-styryl dye destaining as a readout of precise vesicle arrangements and pools in a population of functionally very diverse synapses bears problems. In essence, the approach is 'blind' to many additional processes and confounding factors that operate in the background, from other forms of release to inter-synaptic vesicle exchange. Further, averaging signals over many - functionally very diverse - synapses makes it difficult to distinguish the dynamics of separate vesicle pools within single synapses from a scenario where different kinetics of release originate from different types of synapses with different release probabilities.

The reviewers commented on the revised version of your paper, in essence reiterating the limitations of the approach of bulk imaging of FM de-staining:

(1) The authors sincerely addressed many of the previous concerns, mainly by clarification. The data are consistent with the authors' hypothesis. The pool concept is somewhat similar to that of Richards et al (2000) and Rey et al (2015). The authors further propose that two reserve pools feed vesicles to two readily-releasable pools independently. Unfortunately, the heterogeneity among individual synapses remains a concern as shown in (some of) the raw data (Fig. 3 and supplements). Bulk imaging of FM de-staining does not really measure the fraction of non-stained vesicles, which changes dynamically during stimulation, so that the situation calls for an independent readout of stained and non-stained vesicles. Moreover, direct correspondence between two specific stimulation frequencies (with long stimulation) and vesicle pools is not straightforward. These issues make the experimentally measured pools not well-defined.

(2) The authors' latest round of responses did not alleviate most of my major previous concerns. The additional data now shown in Fig 3 rely on conceptually the same type of bulk measurements and thus suffer from the same limitations as outlined in the earlier review. Moreover, the image of neuronal cultures shown in Fig. 3 might be problematic. It shows very bright staining with large round lumps, which may be indicative of unhealthy cultures.

---

## [Author Response]

The following is the authors’ response to the current reviews.

**Joint Public Review**
This study is concerned with the general question as to how pools of synaptic vesicles are organized in presynaptic terminals to support different types of transmitter release, such as fast synchronous and asynchronous release. To address this issue, the authors employed the classical method of load- ing synaptic vesicle membranes with FM-styryl dyes and assessing dye destaining during repetitive synapse stimulation by live imaging as a readout of the mobilization of vesicles for fusion. Among other 1ndings, the authors provide evidence indicating that there are multiple reserve vesicle pools, that quickly and slowly mobilized reserves do not mix, and that vesicle fusion does not follow a mono-exponential time course, leading to the notion that two separate reserve pools of vesicles - slowly vs. rapidly mobilizing - feed two distinct releasable pools - reluctantly vs. rapidly releasing. These 1ndings are valuable to the 1eld of synapse biology, where the organization of synaptic vesicle pools that support synaptic transmission in different temporal and stimulation regimes has been a focus of intense experimentation and discussion for more than two decades.On the other hand, the present study has limitations, so that the authors’ key conclusions remain incompletely supported by the data, and alternative interpretations of the data remain possible. The approach of using bulk FM-styryl dye destaining as a readout of precise vesicle arrangements and pools in a population of functionally very diverse synapses bears problems. In essence, the approach is ’blind’ to many additional processes and confounding factors that operate in the back- ground, from other forms of release to inter-synaptic vesicle exchange. Further, averaging signals over many - functionally very diverse - synapses makes it diicult to distinguish the dynamics of separate vesicle pools within single synapses from a scenario where different kinetics of release originate from different types of synapses with different release probabilities.

We thank the editors and reviewers for their time and patience, and are happy that they found our results valuable.

We do not have a clear understanding of what the alternative interpretations might be - beyond those already addressed - but would like to, and ask anyone that identifies one to contact us. At present, we believe that the evidence for parallel processing of slowly and quickly mobilized reserve vesicles is compelling and hope that people who are open to the possibility will evaluate the reasoning described within our report.

To be clear, we are not claiming to have tested the further hypothesis - generated by our working model - that quickly and slowly mobilized reserves feed distinct subdivisions of the readily releasable pool. This further hypothesis is consistent with our results, but remains to be tested rigorously.

Beyond that, we have used FM-dye de-staining as a bulk measurement of sub-synaptic events in the sense that we have made no attempt to measure mobilization of isolated individual vesicles. We do not see how this inevitably leaves viable alternative interpretations, but the concern is difficult to evaluate without knowing what the alternatives might be. That is, to our knowledge, the reviewers did not delineate any specific alternatives that are consistent with the full set of results. We can say that the FM-dye technique has been thought to have good resolution at the level of distinguishing between individual synapses since at least Murthy et al. (2001). For our part, we are confident that our analysis in Figure 3 combined with the results in Figures 4-11 shows that many individual presynaptic terminals contain multiple reserve pools that are processed in parallel. We did not use electron microscopy to confirm that all of the punctae analyzed in Figure 3 were indeed single synapses, but the reviewers did not recommend this, and we believe there is already enough published about the spatial distribution of synapses in cell culture to be confident that many of the punctae that are smaller than 1.5 µm were indeed individuals.

Overall, we have attempted to address all of the individual concerns raised by reviewers, and these concerns and our responses are below. Our understanding is that the reviewers were not convinced on every point, but the nature of the concerns was not clear to us. We hope that people who share these concerns will check out our responses and contact us with any further questions or alternative interpretations that may arise.

(1) The authors sincerely addressed many of the previous concerns, mainly by clari1cation. The data are consistent with the authors’ hypothesis. The pool concept is somewhat similar to that of Richards et al (2000) and Rey et al (2015). The authors further propose that two reserve pools feed vesicles to two readily-releasable pools independently.

To clarify further: The possibility that distinct reserve pools feed distinct readily releasable pools is predicted by our working model, and is something that we would like to test in the future, but is not a conclusion of the present study. Instead, in the present study, we tested the prediction that quickly and slowly mobilized reserve vesicles are processed in parallel without making assumptions about the the underlying mechanism (or the number of reserve pools).

Unfortunately, the heterogeneity among individual synapses remains a concern as shown in (some of) the raw data (Fig. 3 and supplements).

We emphasize that we have not attempted to minimize the extensive heterogeneity among synapses, but actually highlight this. In fact, we chose the image in Figure 3 for an example in part because of the lower left region replicated in Figure 3-figure supplement 2 demonstrating extensive heterogeneity along what appears to be a single axon. We are not the first to notice the heterogeneity (see Waters and Smith, 2002), but we do provide a new possible explanation which, if correct, might be important for understanding biological computation (see our Discussion). At the same time, we believe that our evidence for multiple reserve pools with heterogenous properties within individual synapses is compelling. We see no contradiction, and indeed, our conclusion that the ratio of slowly to quickly mobilized varies extensively between synapses can only be correct if individual synapses contain multiple types. We hope that people who are interested in our conclusions will evaluate the evidence and reasoning presented in our report.

Bulk imaging of FM de-staining does not really measure the fraction of non-stained vesicles, which changes dynamically during stimulation, so that the situation calls for an independent readout of stained and non-stained vesicles. Moreover, direct correspondence between two speci1c stimulation frequencies (with long stimulation) and vesicle pools is not straightforward. These issues make the experimentally measured pools not well-de1ned.

We think that the reviewer may be suggesting an alternative scenario where decreases in the fractional rate of FM-dye de-staining seen during 1 Hz stimulation might be caused by a large (4-fold) increase in the total size of the reserve pool that dilutes the stained vesicles by mixing. This scenario is consistent with the results in Figures 2 and 4-7, and initially seems plausible because previous studies have shown that many vesicles are not mobilized, and therefore are not stained, during our standard loading protocol of 100 s at 20 Hz (Harata et al., 2001). However, liberation of this "deep reserve" as an explanation for the decrease in fractional destaining is not compatible with the results in Figures 10-11 that rule out mixing. For example, liberation of the deep reserve would cause fractional destaining to appear equally depressed during subsequent 20 Hz stimulation, and Figure 10 shows that this is not the case. The scenario cannot be rescued by postulating that the subsequent 20 Hz stimulation caused the deep reserve to quickly recapture the liberated vesicles because Figure 11D-E shows that fractional de-staining continues to be depressed at the very beginning of a second 1 Hz train that follows the 20 Hz stimulation.

(2) The authors’ latest round of responses did not alleviate most of my major previous concerns. The additional data now shown in Fig 3 rely on conceptually the same type of bulk measurements and thus suffer from the same limitations as outlined in the earlier review.

We believe that the new evidence in Figure 3 for multiple reserve pools at individual synapses is strong when evaluated in combination with the results in Figures 4-11. We do not, at present, see how the fact that FM-dye destaining is used as a bulk measurement at the sub-synaptic level could undercut our logic.

Moreover, the image of neuronal cultures shown in Fig. 3 might be problematic. It shows very bright staining with large round lumps, which may be indicative of unhealthy cultures.

Unhealthy cultures are not a concern because we used strict quantitative criteria to assess health that are better than we have seen elsewhere (details below). We think the reviewer might be reacting to the way we rendered the image; i.e., as “overexposed”. We did this to highlight the dimmest punctae, which is a key element of the analysis. The same image rendered with less contrast is now displayed in Author response image 1 (3rd panel from left).

**Author response image 1. sa2fig1:** Image to left is a reproduction of the example image in Figure 3, which was the average of 120 time lapse raw data images; scale bar is 20 µm. The second image is a replicate except all 69 punctae that were included in the study are occluded by 1.5 µm × 1.5 µm yellow squares. The third image is another replicate except with a different brightness setting. The rightmost image is one of the raw data images with brightness matched to the third image.

More details (relevance to in vivo is in point 4):

(1) Identifying unhealthy cultures is straightforward with our technique because synapses in unhealthy cultures destain spontaneously. Our criteria for accepting experiments for further analysis was ≤ 1.5 % spontaneous rundown/minute. This is a better way to judge health than we have seen elsewhere because it eliminates subjective decisions, and would be equally applicable for microscopes and imaging software of any quality. For our part, we used a 25X objective with a low numerical aperture and low intensity illumination that allowed us to completely avoid photobleaching. The images will look worse to some compared to when acquired with a higher quality microscope, but the absence of photobleaching is an important benefit because it allowed us to avoid complicated corrections.

(2) Stained areas larger than 1.5 µm across - such as the ones noted by the reviewer - were expressly excluded from our study because they could have been clusters of multiple synapses. The size criteria are detailed in the Legend of Figure 3. Punctae and larger areas that were excluded are the ones that are not occluded by yellow squares in the 2nd image from the left in the "Author resonse image 1", above; at least two of the largest were likely clusters of synapses that were out of focus. Nevertheless, despite being excluded, it is unlikely that the stained areas larger than 1.5 µm in the image in Figure 3 were characteristic of unhealthy synapses because these areas did not de-stain spontaneously, but instead de-stained in response to 1 and 20 Hz electrical stimulation much like the small punctae that were included in the analysis.

(3) Electron microscopy results have shown that individual synapses vary >10-fold in size, so a large range of brightness is expected (Murthy et al., 2001). The large range would either make the brighter punctae and clusters appear to be overexposed in a printed image, or render the dimmer punctae invisible. We have opted to present an image with overall brightness adjusted so that the dimmest punctae are visible. This is appropriate because one of the concerns was that analyzing the dimmest punctae would reveal underlying populations where the rate of fractional destaining was constant. In the end, no evidence for underlying populations emerged, which supports the conclusion that the decreases in fractional destaining occur at individual synapses. Note that adjusting brightness for example images was unavoidable; we used the camera in a range that was far below saturation and, because of this, images presented without adjusting brightness would appear to be completely black.

(4) Primary cell cultures are non-physiological by definition, so the concept of health is intrinsically arbitrary, and relevance to synapses in brains is questioned routinely (quite reasonably, in our opinion). However, the new findings in the present report are that: (1) individual hippocampal synapses contain multiple reserve pools; (2) the reserves remain separate but are not distinguishable by the timing of mobilization when the frequency of stimulation is high; and (3) the reserves are nevertheless processed in parallel even when the frequency of stimulation is high. Of these, finding (1) has been reported previously for other synapse types, but findings (2) and (3) are both without precedent, and finding (3) is not compatible with current concepts. Nevertheless, all three findings were predicted by a model that was developed to explain orthogonal results from studies of intact synapses in ex vivo slices that did not fit with current concepts either, as referenced in the Introduction. Because of this, we think that the parallel processing of quickly and slowly mobilized reserve vesicles likely occurs in individual Schaffer collateral presynaptic terminals in vivo, and is not a cell culture artifact; the alternative would be too much of an unlikely coincidence.

References

Harata N, Pyle JL, Aravanis AM, Mozhayeva M, Kavalali ET & Tsien RW (2001). Limited numbers of recycling vesicles in small CNS nerve terminals: implications for neural signaling and vesicular cycling. Trends in Neurosciences 24, 637–43.

Murthy VN, Schikorski T, Stevens CF & Zhu Y (2001). Inactivity produces increases in neurotransmitter release and synapse size. Neuron 32, 673–82.

Waters J & Smith SJ (2002). Vesicle pool partitioning influences presynaptic diversity and weighting in rat hippocampal synapses. Journal of Physiology 541, 811–23.

The following is the authors’ response to the original reviews.

**Reviewer 1**
Mahfooz et al. investigated the time course of synaptic vesicle fusion of cultured mouse hippocampal synapses using FM-styryl dyes. The major finding is that the FM destaining time course deviates from a mono-exponential function during 1 Hz, but not 20 Hz stimulation. The deviation from a mono-exponential function was also seen during a second stimulus train applied after recovery periods of several minutes, or after depletion of the readily-releasable vesicle pool. Furthermore, this "decreased fractional destaining" was unlikely due to long-term synaptic depression, or incomplete dye clearance. Fractional destaining was enhanced when the dye was loaded with 1 Hz compared with 20 Hz stimulation, suggesting that vesicles recycled during 1 Hz stimulation are predominantly sorted into a rapidly mobilized pool. Finally, they show that20 Hz stimulation does not affect the decrease in fractional destaining induced and recorded during 1 Hz stimulation. Based on these observations, they put forward a model in which slowly and quickly resupplied synaptic vesicles are mobilized in parallel.

The demonstration that FM destaining time courses deviate from single exponentials during 1 Hz stimulation (Figs 2-3) is a starting point used to rule out simple models where vesicles intermix freely and to introduce a mathematical technique for quantifying the extent of the deviations that is essential for the analysis of later experiments, where curve fitting could not be used. We then:

1. show that the deviation from simple models is not caused by depletion of the readily releasable pool, as noted by the reviewer;

2. rule out a number of explanations for the deviation that do not involve reserve pools at all, again as noted;

3. provide affirmative evidence for the presence of multiple reserve pools by labeling them with distinct colors;

4. show that the vesicles within the distinct reserve pools do not intermix even when activity is intense enough to drive destaining with single exponential kinetics.

We believe that the 4th point - documented in Figs 10-11 - is a key element.

Beyond that, we note that our working model arose from previous studies, as referenced in the Introduction, not from the present results. The model did predict the parallel processing of quickly and slowly mobilized reserves, and the present study was designed to test this prediction. In that sense, the evidence in the current study supports our working model, not the other way around.

In any case, most readers in the near term will be more interested in the serial versus parallel question, and less in precisely what the present results mean for evaluating our working model. Because of this, we emphasize that evidence for parallel processing of separate reserve pools depends solely on experimental results within the study, and not on modeling. As a consequence, the evidence will continue to be equally strong even if problems with our working model arise later on (lines 382-386).

We do have additional unpublished evidence for the working model that does not bear directly on the parallel versus serial question. Some of this was removed from an earlier version of the manuscript and some has been newly gathered since the original submission. We will publish the additional evidence at a later point. We decided not to include it in the present manuscript expressly to avoid confusion about the relationship between modeling and the evidence for parallel processing in general.

The paper addresses an interesting question - the relationship between the resupply and release of synaptic vesicles. The study is based on a lot of data of high quality. Most data are solid. However, some of the major conclusions are not well supported by the data. Moreover, it remains unclear how speci1c the findings are to the experimental design.The following points should be addressed:1. Most traces display a decrease in fluorescence intensity before stimulation. Data with a decrease in baseline fluorescence intensity of up to 1.5 % were considered for the analysis (Fig 2-supplement 2). I may have missed it, but were the data corrected for the observed decrease in baseline fluorescence intensity? (In the model shown in Appendix 1 Figure 1, they correct for "rundown"). For instance, are the residuals shown in Fig 2D, E based on corrected data? In case the data would not be corrected for a decrease in baseline fluorescence, would the decay kinetics also deviate from a single exponential after correction?

We did not correct for rundown - as now noted on lines 96-97 - except in the figure in the Appendix, noted by the reviewer, where the uncorrected and corrected time courses are plotted side by side for easy comparison. However, our study includes an analysis showing that correcting for rundown during 1 Hz stimulation would increase - not decrease - the deviation from a single exponential (2 bars in rightmost panel in Fig 2C, and lines 113-116 of Results), so the absence of a correction does not weaken our conclusions.

1. The analysis of "fractional destaining" is not clear to me. How many intervals of which length were chosen and why? For instance, the intervals often differ in length, number and do not cover the complete decay (e.g., Fig 2B).

We calculated fractional destaining from longer intervals at later times because the overall amount of stain was less, meaning signal/noise was less, and scatter was more. We did this because increased scatter at later times could be counteracted by estimating the slope of destaining from longer intervals. An additional benefit is that elongating the later intervals allowed us to plot only 6 bars for 25 min of 1 Hz destaining, which works better visually than 17.

Increasing the interval length for later times is mathematically sound because the key factor causing distortions related to deviations from linearity is not the length of the interval per se but, instead, the fractional destaining over the interval. The fractional destaining is greater at the start of 1Hz stimulation, thus requiring shorter intervals.

It would be possible to choose inappropriately long intervals that would distort estimates of the change in fractional destaining. However, we now include Fig 2-supplement 5 – which includes all 17 1.5 min intervals - to confirm that any distortions after the first interval were minimal. The Appendix predicts a biologically important distortion for the first interval, which would be buried in the noise - we are following this up - but this would underestimate the true deviation from quickly mixing pools, so would not be problematic for the present conclusions.

Sometimes, only the interval right after stimulation onset was considered (e.g., Fig 7, 8).

Figs 7, 8 in the previous version are now Figs 8, 9.

This is appropriate because the goal was to estimate the fractional destaining at the very start, before the quickly mobilized fraction has destained.

How quickly fractional destaining is expected to revert to the lowest value seen after 15 min of 1Hz stimulation in Fig 2 (and elsewhere) depends very much on assumptions - such as the number of reserve pools, etc. We sought to avoid this kind of additional analysis because we are keen to avoid the impression that our main conclusions depend on the specifics of modeling.

How sensitive are the changes in fractional destaining to the choice of the intervals?

Minimally. This can be seen by eye because the magenta lines in Fig 2B fit the data well, but see Fig 2-supplement 5 for a quantitative comparison.

For instance, would fractional destaining be increased if later intervals would have been chosen for the second 20 Hz stimulus in the experiment shown in Fig 9B?

Previous Fig 9B is now Fig 10B.

We cannot be certain, but think it probably would not be different. Neither an increase nor a decrease would be problematic for our conclusions.

More detail: There is not enough data to evaluate this specifically for Fig 10B because the total amount of stain remaining at later intervals is little, meaning signal/noise is low, which causes extensive experimental scatter. However, synapses were even more extensively destained prior to time course c of Figure2-supplement 2C, which nevertheless matches time courses a, b, and d.

I propose fitting all baseline-corrected data with a single and a double-exponential function (as well as single exponential plus line?) and reporting the corresponding time constants (slopes) and amplitudes.

As noted above, we purposefully do not baseline correct data in a way that would make this possible. However, we do include exponential fits when appropriate, in Fig 2D-E, Fig 2- supplement 1, Fig 2-supplement-7, Fig 2-supplement-8, and Fig 12B.

Indeed, the absence of any change in the weighting parameter despite substantial changes for both time constants seen after raising the temperature to 35C (Fig 2-supplement-8 vs Fig 12B) is notable because it suggests that the contents of the reserve pools are not altered by changing temperature, even though vesicle trafficking is accelerated. Fig 2-figure supplement 8 is a supplementary figure because the result is outside the scope of the main point, not because the quality is lower than for other figures.

Beyond that, exponential fits would not be adequate for most of the study because many experiments - including the core experiments in Figs 10-11 - require discontinuous stimulation, such as when we stop stimulating at 1 Hz, rest for minutes, and then start up again at 1 or 20 Hz. And, although widely used, exponentials are non-linear equations after all. Even when they can be used to quantify time courses, the fractional destaining measurement is almost always more informative, in the technical sense, because it avoids complications when estimating the importance of deviations occurring at the two extremes versus deviations in the middle of the time course.

1. Along the same lines, is the average slow time constant indeed around 40 min? (Are the data shown in Fig 2 S7 based on an average?) If this would be the case, I suggest conducting a control experiment with a recording time > 40 min. Would fitting an exponential or a line to baseline data (without stimulation) also give a similar slow component?

Fig 2-figure supplement 7 in the previous version is now Fig 2-figure supplement 8.

First, yes, the time course shown in Fig 2-figure supplement 8 is the mean across preparations. The time courses of the individual preparations were quanti1ed as the median value of the individual ROIs before averaging.

Second, no, fitting baseline data would give an approximately 3-fold greater time constant (i.e., 120 min) because fractional destaining decreases by about 3-fold when we stop stimulating after 25 min of 1 Hz stimulation (i.e., Fig 2C, 3B, and many others).

The key point is that fractional destaining decreases greatly over long trains of 1 Hz stimulation.

For Fig 2, we saw a 2.7+/-0.1-fold decrease before accounting for baseline destaining (lines 106-110), which increased to a 4.4-fold decrease when we did account for baseline destaining (lines 113-116). Overall, the 2.7-fold value is simultaneously a safe minimum boundary, and much greater than the value of 1.0 expected from models where vesicles mix freely.

Note that future studies will show that even the 4.4-fold value is probably an underestimate because 1 Hz stimulation misses a fast component at the very beginning of the time courses, as predicted in the Appendix.

1. How speci1c are the findings to 1 Hz (and 20 Hz) stimulation? From which frequency onward can a decrease in fractional destaining be no longer observed?

Our logic depends only on the premise that we are able to find some frequency where fractional destaining no longer decreases. We knew that 20 Hz was a good place to start because of previous electrophysiological experiments - frequency jumps (Fig 1 of Wesseling and Lo, 2002 and Fig 2C of Garcia-Perez and Wesseling, 2008), and trains of action potentials followed by osmotic shocks (Fig 2A of Garcia-Perez et al., 2008) - showing that 20 Hz stimulation is enough to nearly completely exhaust the readily releasable pool. This is noted in lines 202-203, and Box 2.

would previous stimulation with frequencies <20 Hz interfere with fractional destaining? These control experiments would help assessing how general/speci1c the findings are.

Yes (Figs 4 and 11A at 1 Hz). Also, we have done experiments at 0.1 Hz, which will be published later; some of these were actually removed from an earlier version of the manuscript because the results are primarily relevant to deciding between particular parallel models, and are not relevant to the conclusion of the present study that quickly and slowly mobilized reserves are processed in parallel.

Similarly, a major conclusion of the paper - the parallel mobilization of two vesicle pools - is largely based on these two stimulation frequencies. Can they exclude that mixing between the two pools occurs at other frequencies?

We cannot exclude the possibility of breakdown at a higher frequency, but this would not undercut our conclusions. We do not have plans to try this experiment because: (1) a positive result would be open to concerns about non-physiologically heavy stimulation; and (2) a negative result would be difficult to interpret because of the possibility that the axons cannot follow at higher frequencies.

1. Some information in the methods section is lacking. For instance, which species is the cell culture based on?

Mice from both sexes were used. This is now specified in the Methods.

**Reviewer 2**
By using optical monitoring of synaptic vesicles with FM1-43 at hippocampal synapses, the authors try to show the evidence for two parallel reserve pools of synaptic vesicles, which feed the vesicles to the readily releasable pool. The major strength of the study is the use of a quantitative model, which can be readily testable by experiments: in the course of the study, the authors propose the best vesicle pool model, which fits the experimental data "averaged over synapses" nicely. On the other hand, the weak point of the study comes from the optical method and the data: bulk imaging of vesicle dynamics monitored at each synapse is noisy and the signals vary considerably among synapses. Therefore, the average signals over many synapses may not reflect the vesicle dynamics of two reserve pools within a synapse, but something else, such as the different kinetics of release from multiple synapses with different release probability. Nevertheless, a new framework of two reserve pools offers a testable hypothesis of vesicle dynamics, and the use of single vesicle tracking and EM may allow one to give a de1nitive answer in the future studies Therefore, the study may be of interest to the community of synaptic neurobiology.

1. The current version includes a new figure (Fig 3) showing that the deviations from single pool models seen in populations are caused by deviations occurring at the level of single synapses. The heterogeneity between synapses actually causes population statistics to underestimate - not overestimate - the mean and median size of the deviations at individuals.

We think the new evidence in Fig 3 and supplements is conclusive without follow-on EM of the same punctae, given the substantial body of already published EM on similar cultures. Essentially, the only way to explain the results without invoking multiple reserve pools in individual synapses would be to say that individual synapses ALWAYS come in clumps containing multiple types and are NEVER separated from neighbors by more than 1.5 microns - even when the clumps are separated from each other by 5 microns. There is already clear evidence against this.

1. No new model is proposed here, see the first response to the first reviewer.

2. We are not aware of alternative hypotheses that could account for our results, so cannot evaluate if single vesicle tracking and EM could add meaningful additional support.

1. The existence of non-stained vesicles complicates the interpretation of the data. Because the release by 20 Hz and 1 Hz stimulation do not entirely reflect the release from fast and slow vesicle pools. the estimation of non-stained vesicles using synaptopHluorin (+ba1lomycin) and EPSCs would be helpful to examine fraction of non-stained / stained vesicles over time (with stimulation, the ratio may change dynamically, which may bring complications).

Non-stained vesicles are not a complication, but instead a key element of our logic which is included in the diagrams in Boxes 1 and 2 and Figure 9. That is, quickly and slowly mobilized reserves can be distinguished at 1 Hz precisely because 1 Hz is not intense enough to exhaust the readily releasable pool (Box 2). The corollary is that stained vesicles must be replaced by non-stained vesicles, because otherwise 1 Hz stimulation would exhaust the readily releasable pool. And this is why FM-dyes (plus a beta-cyclodextrin during washing) are ideal for the current questions whereas other techniques, such as electrophysiology or synaptopHluorin imaging are obviously indispensable for other questions, but could not replace the FM-dyes in the current study. This is now noted on lines 86-89.

We are aware that synaptopHluorin + bafilomycin could, in principle, accomplish some of the same goals. However, bafilomycin ended up being toxic when applied for tens of minutes, as it would have to be in our experiments. And, we do not see what critical question is not already answered with strong evidence using FM dyes.

1. Individual synapses show marked differences in the time course of de-staining, suggesting differences in release probability. The averaging of the whole data may reflect "average" behavior of synapses, but for example, bi-exponential time course may reflect high Pr and low Pr synapses, rather than vesicle recruitment.

The authors may comment on this issue.

See newly added Fig 3, and responses above.

1. Some differences are very small (Fig 10, the same amplitude as bleaching time course), and I am not certain if the observed differences are meaningful, given low signal to noise ratio in each synapse.

Fig 10 in the previous version is Fig 11 in the current version.

Even if correct, this would not be problematic because 20 Hz stimulation clearly did not cause fractional destaining to return to the initial value when stimulation was resumed at 1 Hz (compare d and f in Fig 11E). In any case, Figs 2C, 3B, 5B, 7B, and Fig 10-supplement 2A all show that the minimum fractional destaining value during 1 Hz stimulation is about 3-fold greater than during subsequent rest intervals, which is not a small difference. Also, note that Fig 2-supplement 3 shows that photobleaching likely did not play a role.

**Reviewer 3**
Reviewer #3 (Recommendations For The Authors):This study attempts to conceptualize the long-standing question of vesicle pool organization in presynaptic terminals. Authors used classical FM dye release experiments to support a hypothesis that rapidly and slowly releasing vesicles are mobilized in parallel without intermixing. This modular model is also supported indirectly by the authors’ recent findings of molecular links that connect a subset of vesicles in linear chains (published elsewhere).

Our study should be seen as a test of the hypothesis that quickly and slowly mobilized reserves are processed in parallel. The evidence is independent of any modeling, and would continue to be equally strong if our working model turns out to be incorrect (lines 382-386).

The scope of the original model was limited by a number of caveats. The main concerns included a limited data set measured in bulk from a highly heterogeneous synapse population, and a complex interrelationship between vesicle mobilization and the bulk FM dye de-staining kinetics. The second major limitation was measurements being performed at room temperature, which inhibits or alters a number of critical synaptic processes that are being modeled. This includes the efficiency of exo/endocytosis coupling, vesicle mobility and release site refractory period, which are stimulus- and temperature-dependent, but were not accounted for in the original model.

The present study contains experiments at body temperature (Fig 12 and Fig 12-supplement 1 in the current version) and analyses of individual synapses (especially Fig 3 in the current version). To our knowledge all results are consistent with everything that is known about the efficiency of exo/endocytosis coupling, vesicle mobility and release site refractory periods.

The authors made strong efforts to address previous concerns. However, the main conceptual point, i.e. linking the bulk FM dye de-staining kinetics with precise arrangement of vesicle pools, is not well supported and is generally highly problematic because it ignores many additional processes and confounding factors.For example, vesicle exchange between neighboring synapses constitutes from 15% to over 50% of total recycling vesicle population, and therefore is a major contributing factor to FM dye loss/redistribution, but is not considered in this study. Additionally, this vesicle exchange process undergoes calcium/activity-dependent changes, contributing to difficulty in interpreting the current experiments comparing FM de-staining at different stimulation frequencies.

We do not see how exchange of vesicles between synapses could be a problem for our logic, so cannot evaluate this without a more detailed description of the concern. Instead, our results rule out random inter-synaptic exchange between quickly and slowly mobilized reserve pools because this would show up in our assays as mixing, which does not occur. We think there are three remaining possibilities:

1. vesicles are exchanged primarily between quickly mobilized reserve pools

2. vesicles are exchanged primarily between slowly mobilized reserve pools

3. vesicles in quickly mobilized reserve pools are targeted to quickly mobilized reserve pools in other synapses and vesicles in slowly mobilized reserve pools are targeted to slowly mobilized reserve pools in other synapses.

It would be interesting to know which of these is correct, but this is outside the scope of thecurrent study.

Moreover, other forms of release, such as asynchronous release, contribute a large fraction of released vesicles, but are not factored in. Asynchronous release varies widely in synapse population from 0.1 to >0.4 of synchronous release, but is entirely ignored. Spontaneous release may also contribute to FM dye loss over extended 25min recordings used.

Spontaneous release and asynchronous release are not caveats.

First, spontaneous: We suspect that spontaneous release contributes to the background destaining rate, but this is 3-fold slower than the minimum during 1 Hz stimulation on average (Figs 2C, 3C, 5B etc), so we know that the slowly mobilized reserve is mobilized by low frequency trains of action potentials (lines 410-412). Note that a different outcome - where the rate of destaining decreased to a very low level during long trains of 1 Hz stimulation - would not have been consistent with the idea that slowly mobilized vesicles are only released spontaneously because the remaining fluorescence can always be destained rapidly by increasing the stimulation intensity to 20 Hz (e.g., see examples in Fig 3).

Second, asynchronous: We know that slowly mobilized reserves must be released synchronously at 35C because the asynchronous component is eliminated at this temperature (Huson et al., 2019), without altering the quantity of slowly mobilized reserves that are mobilized by 1 Hz stimulation (lines 350-360 of Results, and 445-452 of Discussion; we can confirm from our own unpublished experiments that the disappearance of asynchronous release at 35C is a robust phenomenon in these cell cultures). Asynchronous release of slowly mobilized vesicles might occur at room temperature, but this would not argue against the conclusion that slowly mobilized vesicles are processed in parallel with quickly mobilized.

Speci1c comments:Points 1-4 are already addressed above.1. The notion of the chained vesicles is somewhat confusing: how does the "first" vesicle located at the plasma membrane/release site get released if it is attached to the chain? Wouldn’t this "first" vesicle be non-immediately releasable since it must first be liberated? Since all vesicles shown in the Figure 1 have chains attached to them, what vesicle population then give rise to sub-millisecond release?

This is not a concern that is relevant to the present study because none of the conclusions rely on the model in any way (see Introduction, and lines 382-386 of the Discussion). Beyond that: We previously published clear evidence that docked vesicles are tethered to non-docked vesicles (Figure 8 of Wesseling et al., 2019). We see no reason to suspect that a tether to an internal vesicle would prevent the docked vesicle from priming for release.

1. Model: For fitting de-staining during 20 Hz stimulation, authors state that it was necessary to allow >5-fold Facilitation. This seems to be non-physiologically relevant, since previous studies found only very mild facilitation at room temperature (typically below a factor of 1.5-2.0) and the authors themselves state that, at most, a 1.3 fold facilitation was found.

If the 1.3-fold facilitation estimate comes from us, it must have been in a different context.

Most estimates of facilitation that are published are heavily convolved with simultaneous depression, and there is additionally a saturation mechanism for readily releasable vesicles with high release probability that is not widely known (Garcia-Perez and Wesseling, 2008). The standard method for eliminating the depression is to lower the probability of release by lowering extracellular [Ca2+], which additionally relieves occlusion by the saturation mechanism. And, lowering [Ca2+] uncovers an enormous amount facilitation at synapses in hippocampal cell culture. For example, see Figure 2B of Stevens and Wesseling (1999), which shows a 7-fold enhancement during 9 Hz stimulation, and Figure 3 of the same study, which shows a linear relationship with frequency. Taken together these two results suggest 15-fold enhancement during 20 Hz stimulation, which far exceeds the 5-fold value needed at inefficient release sites to make our working model 1t the FM-dye destaining results.

References

Garcia-Perez E, Lo DC & Wesseling JF (2008). Kinetic isolation of a slowly recovering component of short-term depression during exhaustive use at excitatory hippocampal synapses. Journal of Neurophysiology 100, 781–95.

Garcia-Perez E & Wesseling JF (2008). Augmentation controls the fast rebound from depression at excitatory hippocampal synapses. Journal of Neurophysiology 99, 1770–86.

Huson V, van Boven MA, Stuefer A, Verhage M & Cornelisse LN (2019). Synaptotagmin-1 enables frequency coding by suppressing asynchronous release in a temperature dependent manner. Scienti1c reports 9, 11341.

Stevens CF & Wesseling JF (1999). Augmentation is a potentiation of the exocytotic process. Neuron 22, 139–46.

Wesseling JF & Lo DC (2002). Limit on the role of activity in controlling the release-ready supply of synaptic vesicles. Journal of Neuroscience 22, 9708–20.

Wesseling JF, Phan S, Bushong EA, Siksou L, Marty S, Pérez-Otaño I & Ellisman M (2019). Sparse force-bearing bridges between neighboring synaptic vesicles. Brain Structure and Function 224, 3263–3276.